# Exploring the Uncertainty Properties of Neural Networks' Implicit Priors in the Infinite-Width Limit

**Ben Adlam**[*†]    **Jaehoon Lee** [*]    **Lechao Xiao** [*]    **Jeffrey Pennington**    **Jasper Snoek**
Google Brain
{adlam, jaehlee, xlc, jpennin, jsnoek}@google.com

## Abstract

Modern deep learning models have achieved great success in predictive accuracy for many data modalities. However, their application to many real-world tasks is restricted by poor uncertainty estimates, such as overconfidence on out-of-distribution (OOD) data and ungraceful failing under distributional shift. Previous benchmarks have found that ensembles of neural networks (NNs) are typically the best calibrated models on OOD data. Inspired by this, we leverage recent theoretical advances that characterize the function-space prior of an infinitely-wide NN as a Gaussian process, termed the neural network Gaussian process (NNGP). We use the NNGP with a softmax link function to build a probabilistic model for multi-class classification and marginalize over the latent Gaussian outputs to sample from the posterior. This gives us a better understanding of the implicit prior NNs place on function space and allows a direct comparison of the calibration of the NNGP and its finite-width analogue. We also examine the calibration of previous approaches to classification with the NNGP, which treat classification problems as regression to the one-hot labels. In this case the Bayesian posterior is exact, and we compare several heuristics to generate a categorical distribution over classes. We find these methods are well calibrated under distributional shift. Finally, we consider an infinite-width final layer in conjunction with a pre-trained embedding. This replicates the important practical use case of transfer learning and allows scaling to significantly larger datasets. As well as achieving competitive predictive accuracy, this approach is better calibrated than its finite width analogue.

## 1 Introduction

Large, neural network (NN) based models have demonstrated remarkable predictive performance on test data drawn from the same distribution as their training data, but the demands of real world applications often require stringent levels of robustness to novel, changing, or shifted distributions. Specifically, we might ask that our models are calibrated. Aggregated over many predictions, well calibrated models report confidences that are consistent with measured performance. The Brier score (BS), expected calibration error (ECE), and negative log-likelihood (NLL) are common measurements of calibration (Brier, 1950; Naeini et al., 2015; Gneiting & Raftery, 2007).

Empirically, there are many concerning findings about the calibration of deep learning techniques, particularly on *out-of-distribution* (OOD) data whose distribution differs from that of the training data. For example, MacKay (1992) showed that non-Bayesian NNs are overconfident away from the training data and Hein et al. (2019) confirmed this theoretically and empirically for deep NNs that use ReLU. For in-distribution data, *post-hoc* calibration techniques such as temperature scaling tuned on a validation set (Platt et al., 1999; Guo et al., 2017) often give excellent results; however, such methods have not been found to be robust on shifted data and indeed sometimes even reduce calibration on such data (Ovadia et al., 2019). Thus finding ways to detect or build models that produce reliable probabilities when making predictions on OOD data is a key challenge.

---

[*]Authors contributed equally to this work. [†]Work done as a member of the Google AI Residency program (https://g.co/airesidency).

Sometimes data can be only slightly OOD or can shift from the training distribution gradually over time. This is called *dataset shift* (Quionero-Candela et al., 2009) and is important in practice for models dealing with seasonality effects, for example. While perfect calibration under arbitrary distributional shift is impossible, simulating plausible kinds of distributional shift that may occur in practice at different intensities can be a useful tool for evaluating the calibration of existing models. A recently proposed benchmark takes this approach (Ovadia et al., 2019). For example, using several kinds of common image corruptions applied at various intensities, the authors observed the degradation in accuracy expected of models trained only on clean images (Hendrycks & Dietterich, 2019; Mu & Gilmer, 2019), but also saw very different levels of calibration, with deep ensembles (Lakshminarayanan et al., 2017) proving the best.

**Calibration metrics.** There are many common metrics used to evaluate the calibration of a model. Probably most common is the negative log-likelihood (NLL), which is a proper scoring rule and whose units, nats, are a coherent unit for information entropy. The Brier score (BS) from Brier (1950) is also a proper scoring rule, but its scale is harder to interpret and its use of the squared loss makes it sensitive to in/frequent events. We also consider the expected calibration error (ECE) of Naeini et al. (2015), which is not a proper scoring rule. In particular, uniformly random predictions on a balanced, classification dataset will produce an ECE of 0. However, since it measures the absolute difference between predicted and observed probabilities, the scale of ECE is easier to interpret. Finally, since the previous metrics require labels, we consider the average confidence and entropy on OOD data, where the confidence is the probability assigned to the most likely class and the entropy is the entropy of the predicted distribution over labels. Ovadia et al. (2019) has more discussion on calibration metrics.

**Bridging Bayesian Learning and Neural Networks.** In principle, Bayesian methods provide a promising way to tackle calibration, allowing us to define models with and infer under specific aleatory and epistemic uncertainty. Typically, the datasets on which deep learning has proven successful have high SNR (Signal-to-Noise Ratio), meaning epistemic uncertainty is dominant and model averaging is crucial because our overparameterized models are not determined by the training data. Indeed Wilson (2020) argues that ensembles are a kind of Bayesian model average.

Ongoing theoretical work has built a bridge between NNs and Bayesian methods (Neal, 1994a; Lee et al., 2018; Matthews et al., 2018b), by identifying NNs as converging to Gaussian processes in the limit of very large width. Specifically, the neural network Gaussian process (NNGP) describes the prior on function space that is realized by an i.i.d. prior over the parameters. The function space prior is a GP with a specific kernel that is defined recursively with respect to the layers. While the many heuristics used in training NNs may obfuscate the issue, little is known theoretically about the uncertainty properties implied by even basic architectures and initializations of NNs. Indeed theoretically understanding overparameterized NNs is a major open problem. With the NNGP prior in hand, it is possible to disambiguate between the uncertainty properties of the NN prior and those due to the specific optimization decisions by performing Bayesian inference. Moreover, it is only in this infinite-width limit that the posterior of a Bayesian NN can be computed exactly.

## 1.1 Summary of contributions

This work is the first extensive evaluation of the uncertainty properties of infinite-width NNs. Unlike previous work, we construct a valid probabilistic model for classification tasks using the NNGP, *i.e.* a label's prediction is always a categorical distribution. We perform neural network Gaussian process classification (NNGP-C) using a softmax link function to exactly mirror NNs used in practice. Note that prior work on the NNGP has also used a softmax link-function, but considered approximate inference using inducing points on MNIST (Garriga-Alonso et al., 2019). We perform a detailed comparison of NNGP-C against its corresponding NN on clean, OOD, and shifted test data and find NNGP-C to be significantly better calibrated and more accurate than the NN.

Next, we evaluate the calibration of neural network Gaussian process regression (NNGP-R) on both UCI regression problems and classification on CIFAR10. As the posterior of NNGP-R is a multivariate normal and so not a categorical distribution, a heuristic must be used to calculate confidences for classification problems. On the full benchmark of Ovadia et al. (2019), we compare several such heuristics, and against the standard RBF kernel and ensemble methods. We find the calibration of NNGP-R to be competitive with the best results reported in Ovadia et al. (2019). However in the process of preparing our findings for publication, newer strong baselines have been

reported that we do not compare against.[1]

Finally, we consider NNs whose last layer only is infinite by taking a pre-trained embedding and using an infinite-width final layer (abbreviated as NNGP-LL). This mirrors an important practical use case for practitioners unable to retrain large models from scratch, but looking to adapt one to their particular data, which could have relatively few samples. We compare the calibration of NNGP-LL with a multi-layer FC network, fine-tuning of the whole network, and the gold standard ensemble method, and again we find NNGP-LL to have competitive calibration. While NNGP-LL is not a principled Bayesian method, it allows scaling to much larger datasets and potential synergy with recent advances in un- and semi-supervised learning due to the pre-training of the embedding, making this method applicable to many real-world uncertainty critical applications like medical imaging.

In addition, code for some of our experiments and precomputed kernels are also publicly available for any further analysis by the research community.[2]

## 2 BACKGROUND

Neal (1994b) identified the connection between infinite-width NNs and Gaussian processes, showing that the outputs of a randomly initialized one-hidden layer NN converge to a Gaussian process as the number of units in the layer approaches infinity. Let $z_i^l(x)$ describe the $i^{\text{th}}$ pre-activation following a linear transformation in the $l^{\text{th}}$ layer of a NN. At initialization, the parameters of the NN are independent and random, so the central-limit theorem can be used to show that the pre-activations become Gaussian with zero mean and a covariance matrix $\mathcal{K}(x, x') = \mathbb{E}[z_i^l(x) z_i^l(x')]$.[3]

Knowing the distributions of the outputs, one can apply Bayes theorem to compute the posterior distribution for new observations, which we detail in Sec. 3 for classification and Sec. 4 for regression. Moreover, when the parameters evolve under gradient flow to minimize a squared loss, the output distribution[4] remains a GP but with a different kernel called the neural tangent kernel (NTK) (Jacot et al., 2018; Lee et al., 2019). This allows us to derive the exact formula for *ensemble training* of infinite width networks; see Sec. A for more details. While we also analyzed the uncertainty properties of the NTK in our experiments, we observed that the results mirror those of the NNGP.

These observations have recently been significantly generalized to include NNs with more than one hidden layer (Lee et al., 2018; Matthews et al., 2018b) and with a variety of architectures, including weight-sharing (CNNs) (Xiao et al., 2018; Novak et al., 2019b; Garriga-Alonso et al., 2019), skip-connections, dropout (Schoenholz et al., 2017; Pretorius et al., 2019), batch norm (Yang et al., 2019), weight-tying (RNNs)(Yang, 2019), self-attention (Hron et al., 2020), graphical networks (Du et al., 2019), and others. In this work, we focus on FC and CNN-Vec NNGPs, whose kernels are derived from fully-connected networks and convolution networks without pooling respectively (see Sec. A for precise definitions). When it is required, we prepend FC- or C- to the NNGP to distinguish between these two variants. We use the Neural Tangents library of Novak et al. (2019a) to automate the transformation of finite-width NNs to their corresponding Gaussian processes.

Another line of work has connected NNs and GPs by combining them in a single model. Hinton & Salakhutdinov (2008) trained a deep generative model and fed its output in to a GP for discriminative tasks. Wilson et al. (2016b) consider a similar approach, creating a GP whose kernel is given as the composition of a nonlinear mapping (*e.g.* a NN) and a spectral kernel. They scale this approach to larger datasets in Wilson et al. (2016a) using variational inference, but Tran et al. (2018) found this method to be poorly calibrated.

## 3 FULL BAYESIAN TREATMENT OF CLASSIFICATION WITH NEURAL KERNELS

It is common to interpret the logits of a NN once mapped through a softmax as a categorical distribution over the labels for each point. Indeed cross entropy loss is sometimes motivated as the KL divergence between the predicted distribution and the observed label. Similarly, while the

---

[1] https://github.com/google/uncertainty-baselines

[2] https://github.com/google-research/google-research/tree/master/infinite_uncertainty

[3] Note that $\mathcal{K}$ is independent of $i$, due to the independence between $z_i^l$ and $z_{i'}^l$ for $i \neq i'$. In other words, the covariances of the GP have a Kronecker factorization $\mathcal{K} \otimes \text{I}_c$, where $c$ is the number of classes.

[4] The source of randomness comes from random initialization.

initialization scheme used for a NN's parameters is often chosen for optimization reasons, it can also be thought of as a prior. This implicit prior over functions and over the distribution of labels has effects, despite the decision of most common training algorithms in deep learning to forgo explicitly trying to find its posterior. In this section, we take seriously this implicit prior and utilize the simple characterization it has over logits in the infinite width limit to define a probabilistic model for mutli-class classification as

$$(y|x) \sim \text{softmax}(f(x)), \text{ where } f \sim \mathcal{GP}(\mathbf{0}, \mathcal{K}) \tag{1}$$

and $\mathcal{K}$ is the NNGP kernel. In words, Eq. (1) states that the latent space is a Gaussian process with mean 0 and kernel function $\mathcal{K}$ and the softmax function is used to transform the latent space into a distribution over classes for each point $x$. If this prior is a correct model for the data generation process, then the posterior is optimal for inference. Thus, by avoiding heuristic approaches to inference, we are able to directly evaluate the statistical properties of the prior. Then by comparing this to more standard gradient-based training methods, we can understand their effect on calibration.

For training data $(\mathcal{X}, \mathcal{Y})$, the posterior on a test point $x$ can be found by marginalizing out the latent space. Denote the latent vector for the whole training data with $F := f(\mathcal{X})$ and the test point with $f := f(x)$, and then let $\mathbf{F}$ denote their concatenation. Then the posterior for the test label can be calculated by integrating over the posterior on the latent space of the test point:

$$p(y|x, \mathcal{X}, \mathcal{Y}) = \int \text{softmax}(f)p(f|x, \mathcal{X}, \mathcal{Y}) \, df. \tag{2}$$

Note that $p(f|x, \mathcal{X}, \mathcal{Y})$ is a marginal of the latent vectors' posterior distribution on all points, *i.e.*

$$p(\mathbf{F}|x, \mathcal{X}, \mathcal{Y}) = \frac{p(\mathcal{Y}|F, x, \mathcal{X})p(\mathbf{F}|x, \mathcal{X})}{p(\mathcal{Y}|x, \mathcal{X})}, \tag{3}$$

which, after neglecting the normalizing denominator, is proportional to the prior in the latent space and the probabilities under the softmax link-function, $\mathcal{N}(\mathbf{F}; \mathbf{0}, \mathcal{K}) \prod_i \text{softmax}(F_i)_{\mathcal{Y}_i}$. See Williams & Rasmussen (2006) for more details. We generate samples from the joint posterior distribution of $f$ and $F$ using elliptical slice sampling (ESS) (Murray et al., 2010).

Note that the latent space dimension in the datasets we consider is substantial, which makes inference with ESS computationally intensive. Specifically, each step of ESS requires sampling from the prior and thus a matrix multiplication of the Cholesky decomposition of the kernel matrix (whose computational time is cubic in the dataset size but can be precomputed once), and the convergence times are longer for larger systems. Moreover, while ESS is hyperparameter free, we found its convergence is affected by the kernel hyperparameters. We release open-sourced JAX based implementation of GPC based on ESS described here.

Given these computational constraints on both tuning hyperparameters and running ESS, we focus our attention on FC and simple CNN-Vec kernels, which are much faster to compute than pooling based kernels.[5] Since we wish to make like-for-like comparisons between the Gaussian process and the corresponding NN, this restricts us to simplistic NN architectures that are far from state-of-the-art. We find this acceptable as the goal of this section is relative comparison and not to obtain optimal absolute performance.[6]

To tune the kernel hyperparameters (weight and bias standard deviations, the activation function, the depth, and the kernel jitter) we used the Google Vizier service (Golovin et al., 2017b) with a budget of 250 trials and selected the setting with the best log-likelihood on a validation set. We use the same hyperparamters for the NN to make a direct comparison of the prior. The additional hyperparameters required for the NN, like width and training time, were also tuned using Vizier. We also compared against the NN performance when all its hyperparameters are tuned, and found the accuracy of the NN improved but the calibration results were broadly similar. See the supplement for additional details. Note that while optimizing hyperparamters based on a validation set likelihood is nonstandard for Bayesian methods, we do so here to make the fairest comparison to NNs, whose hyperparamters are normally optimized in this way.

---

[5]For reference, the relatively efficient Myrtle kernel of Shankar et al. (2020) takes approximately 12,000 V100 GPU hours to compute for CIFAR10 and CIFAR10-C for a single hyperparameter configuration.

[6]Making a fair comparison between finite versus infinite-width network is subtle, requiring careful thought controlling for architecture and training methods (Lee et al., 2020).

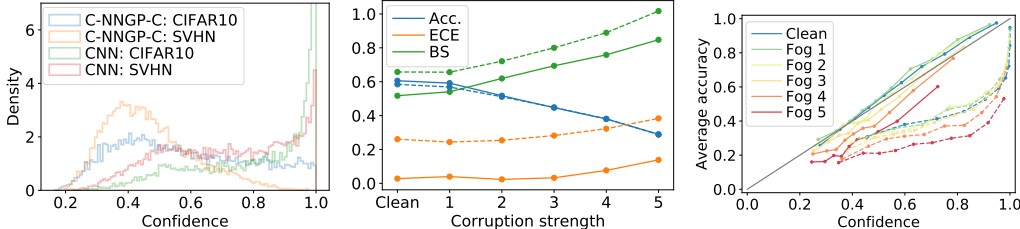

Figure 1: Investigating the calibration of Gaussian process classification with CNN-GP kernels. (**left**) Histogram of the confidence of the posterior distribution for each test point. We compare the C-NNGP-C and a finite width CNN on an in-distribution test set (CIFAR10) and an OOD test set (SVHN). C-NNGP-C shows lower confidence and higher entropy (see Fig. S2) on both test sets compared to the CNN, which has very high confidence on many points as indicated by the spike around 1. On the clean data the CNN's overconfidence hurts its calibration, as it achieves worse ECE and BS than C-NNGP-C. (**middle**) The performance of both models, C-NNGP-C is solid and CNN dashed, under increasing distributional shift given by the CIFAR10 fog corruption from Hendrycks & Dietterich (2019). The accuracy of the CNN and C-NNGP-C are comparable as the shift intensity increases, but C-NNGP-C remains better calibrated throughout. (**right**) We bin the test set into ten bins sorted by confidence, and we plot mean confidence against mean accuracy. C-NNGP-C remains closer to $x = y$ than the CNN, which shifts toward increasing overconfidence.

Table 1: Comparing NNs and against the equivalent NNGP-C on CIFAR-10 and evaluated on several test sets. We observe that the NNGP-C outperforms its parametric NN counterpart on *every* metric. We see particularly significant improvements in ECE and NLL, implying that NNGP-C is considerably better calibrated. For complete results on other intensities of Fog corruption see Table S1.

| Data | Metric | FC-NN | FC-NNGP-C | CNN | C-NNGP-C |
|------|--------|-------|-----------|-----|----------|
| CIFAR10 | ECE | 0.209 | **0.072** | 0.283 | **0.031** |
| | Brier Score | 0.711 | **0.629** | 0.685 | **0.519** |
| | Accuracy | 0.486 | **0.518** | 0.576 | **0.609** |
| | NLL | 17383 | **14007** | 21786 | **11215** |
| Fog 1 | ECE | 0.178 | **0.098** | 0.271 | **0.042** |
| | Brier Score | 0.707 | **0.655** | 0.685 | **0.542** |
| | Accuracy | 0.471 | **0.496** | 0.561 | **0.590** |
| | NLL | 16702 | **14671** | 19716 | **11755** |
| Fog 5 | ECE | 0.279 | **0.057** | 0.420 | **0.134** |
| | Brier Score | 0.961 | **0.847** | 1.052 | **0.846** |
| | Accuracy | 0.241 | **0.250** | 0.287 | **0.289** |
| | NLL | 26345 | **20694** | 33891 | **22014** |
| SVHN | Mean Confidence | 0.537 | 0.335 | 0.718 | 0.463 |
| | Entropy | 1.230 | 1.840 | 0.733 | 1.403 |
| CIFAR100 | Mean Confidence | 0.651 | 0.398 | 0.812 | 0.474 |
| | Entropy | 0.944 | 1.663 | 0.493 | 1.420 |

In our experiments, we compare NNGP-C and a finite NN of the same architecture on in-distribution, shifted, and OOD test data when both models are trained using the standard CIFAR10 training set. We consider CIFAR100, which contains classes not present in CIFAR10, and SVHN, which consists of images of house numbers, as two examples of OOD data. The five levels of fog corruptions from Hendrycks & Dietterich (2019), which distorts the CIFAR10 test set with a fog-like effect, are used as increasing amounts of distributional shift as in Ovadia et al. (2019). Our main findings for NNGP-C, summarized in Fig. 1 and Table 1, show that NNGP-C is well calibrated and outperforms the corresponding NN. This indicates that, rather than the prior induced by the random initialization scheme, the MAP-based training of the NN is partly responsible for its poor calibration.

## 4 REGRESSION WITH THE NNGP

As observed in Sec. 3, inference with NNGP-C is challenging as the posterior is intractable. In this section, we consider Gaussian process regression using the NNGP (abbreviated as NNGP-R), which

is defined by the model

$$(y|x) \sim f(x) + \varepsilon, \text{ where } f \sim \mathcal{GP}(\mathbf{0}, \mathcal{K}), \tag{4}$$

where $\mathcal{K}$ is the NNGP kernel and $\varepsilon \sim \mathcal{N}(0, \sigma_\epsilon^2)$ is an independent noise term. One major advantage of NNGP-R is that the posterior is analytically tractable. The posterior at a test point $x$ has a Gaussian distribution with mean and variance given by

$$\mu(x) = \mathcal{K}(x, \mathcal{X})\mathcal{K}_\epsilon(\mathcal{X}, \mathcal{X})^{-1}\mathcal{Y} \quad \text{and} \quad \sigma^2(x) = \mathcal{K}(x, x) - \mathcal{K}(x, \mathcal{X})\mathcal{K}_\epsilon(\mathcal{X}, \mathcal{X})^{-1}\mathcal{K}(\mathcal{X}, x), \tag{5}$$

where $(\mathcal{X}, \mathcal{Y})$ is the training set of inputs and targets respectively and $\mathcal{K}_\epsilon \equiv \mathcal{K} + \sigma_\epsilon^2 \mathbf{I}$. Since $\mathcal{K}$ has a Kronecker factorization, the complexity of inference is $c\mathcal{O}(|\mathcal{X}|^3)$ rather than $\mathcal{O}(c^3|\mathcal{X}|^3)$. For regression problems where $y \in \mathbb{R}^d$, the variance describes the model's uncertainty about the test point. Note that while this avoids the difficulties of approximate inference methods like MCMC, computation of the kernel inverse means running time scales cubically with the dataset size.

### 4.1 BENCHMARK ON UCI DATASETS

We perform non-linear regression experiments proposed by Hernández-Lobato & Adams (2015), which is a standard benchmark for evaluating uncertainty of Bayesian NNs. We use all datasets except for `Protein` and `Year`. Each dataset is split into 20 train/test folds. We report our result in Table 2, comparing against the following strong baselines: Probabilistic BackPropagation with the Matrix Variate Gaussian distribution (PBP-MV) (Sun et al., 2017), Monte-Carlo Dropout (Gal & Ghahramani, 2016) evaluated with hyperparameter tuning as done in Mukhoti et al. (2018) and Deep Ensembles (Lakshminarayanan et al., 2017). We also evaluated a standard GP with RBF kernel $K_{\text{RBF}}(x, x') = \beta \exp\left(-\gamma||x - x'||^2\right)$ for comparison.

Instead of maximizing train NLL for model selection, we performed hyperparameter search on a validation set (we further split the training set so that overall train/valid/test split is 80/10/10), as commonly done in NN model selection and in the BNN context applied in Mukhoti et al. (2018). For NNGP-R, the following hyperparameters were considered. The activation function is chosen from (ReLU, Erf), number of hidden layers among [1, 4, 16], $\sigma_w^2$ from [1, 2, 4], $\sigma_b^2$ from [0., 0.09, 1.0], readout layer weight and bias variance are chosen either same as body or $(\sigma_w^2, \sigma_b^2) = (1, 0)$. For the GP with RBF kernel we evaluated $\gamma \in \{10^k : k \in [-5, -4, ..., 3]\}$ and $\beta \in \{10^k : k \in [-3, -4, ..., 3]\}$. For both experiments, $\sigma_\varepsilon \in$ `np.logspace(-6, 4, 20)`.

We found that NNGP-R can outperform and remain competitive with existing methods in terms of both root-mean-squared-error (RMSE) and negative-log-likelihood (NLL). In Table 2, we observe that NNGP-R achieves the lowest RMSE on the majority (5/8) of the datasets and competitive NLL. When using the NTK as the GP's kernel instead, we saw broadly similar results.

### 4.2 CLASSIFICATION AS REGRESSION

Formulating classification as regression often leads to good results, despite being less principled (Rifkin et al., 2003; Rifkin & Klautau, 2004). By doing so, we can compare exact inference for GPs to trained NNs on well-studied image classification tasks. Recently, various studies of infinite NNs have considered classification as regression tasks, treating the one-hot labels as independent regression targets (*e.g.* Lee et al. (2018); Novak et al. (2019b); Garriga-Alonso et al. (2019)). Predictions are then obtained as the argmax of the mean in Eq. (5), *i.e.* $\arg\max_k \mu(x)_k$.[7]

However, this approach does not provide confidences corresponding to the predictions. Note that the posterior gives support to all of $\mathbb{R}^d$, including points that are known to be impossible. Thus, a heuristic is required to extract meaningful uncertainty estimates from the posterior Eq. (5), even though these confidences will not correspond to the Bayesian posterior of any model.

Following Albert & Chib (1993); Girolami & Rogers (2006), we produce a categorical distribution for each test point $x$, denoted $p_x$, by defining

$$p_x(i) := \mathbb{P}[y_i = \max\{y_1, \ldots, y_d\}] = \int \mathbf{1}(i = \operatorname{argmax}_j y_j) \prod_{k=1}^{K} p(y_k|x, \mathcal{X}, \mathcal{Y}) dy, \tag{6}$$

---

[7]While training NNs with MSE loss is more challenging, peak performance can be competitive with cross-entropy loss (Lewkowycz et al., 2020; Hui & Belkin, 2020; Lee et al., 2020).

Table 2: Result for regression benchmark on UCI Datasets. Note $\pm x$ reports the standard error around estimated mean for 20 splits. We compare to strong baselines: PMP-MV (Sun et al., 2017), MC-Dropout from Mukhoti et al. (2018) and Deep Ensembles (Lakshminarayanan et al., 2017)

### Average RMSE Test Performance

| Dataset | $(m, d)$ | PBP-MV | Dropout | Ensembles | RBF | FC-NNGP-R |
|---|---|---|---|---|---|---|
| Boston Housing | (506, 13) | $3.11 \pm 0.15$ | $\mathbf{2.90 \pm 0.18}$ | $3.28 \pm 1.00$ | $3.24 \pm 0.21$ | $3.07 \pm 0.24$ |
| Concrete Strength | (1030, 8) | $5.08 \pm 0.14$ | $\mathbf{4.82 \pm 0.16}$ | $6.03 \pm 0.58$ | $5.63 \pm 0.24$ | $5.25 \pm 0.20$ |
| Energy Efficiency | (768, 8) | $\mathbf{0.45 \pm 0.01}$ | $0.54 \pm 0.06$ | $2.09 \pm 0.29$ | $0.50 \pm 0.01$ | $0.57 \pm 0.02$ |
| Kin8nm | (8192, 8) | $\mathbf{0.07 \pm 0.00}$ | $0.08 \pm 0.00$ | $0.09 \pm 0.00$ | $\mathbf{0.07 \pm 0.00}$ | $\mathbf{0.07 \pm 0.00}$ |
| Naval Propulsion | (11934, 16) | $\mathbf{0.00 \pm 0.00}$ | $\mathbf{0.00 \pm 0.00}$ | $\mathbf{0.00 \pm 0.00}$ | $\mathbf{0.00 \pm 0.00}$ | $\mathbf{0.00 \pm 0.00}$ |
| Power Plant | (9568, 4) | $3.91 \pm 0.04$ | $4.01 \pm 0.04$ | $4.11 \pm 0.17$ | $3.82 \pm 0.04$ | $\mathbf{3.61 \pm 0.04}$ |
| Wine Quality Red | (1588, 11) | $0.64 \pm 0.01$ | $0.62 \pm 0.01$ | $0.64 \pm 0.04$ | $0.64 \pm 0.01$ | $\mathbf{0.57 \pm 0.01}$ |
| Yacht Hydrodynamics | (308, 6) | $0.81 \pm 0.06$ | $0.67 \pm 0.05$ | $1.58 \pm 0.48$ | $0.60 \pm 0.07$ | $\mathbf{0.41 \pm 0.04}$ |

### Average Negative Log-Likelihood Test Performance

| Dataset | $(m, d)$ | PBP-MV | Dropout | Ensembles | RBF | FC-NNGP-R |
|---|---|---|---|---|---|---|
| Boston Housing | (506, 13) | $2.54 \pm 0.08$ | $\mathbf{2.40 \pm 0.04}$ | $2.41 \pm 0.25$ | $2.63 \pm 0.09$ | $2.65 \pm 0.13$ |
| Concrete Strength | (1030, 8) | $3.04 \pm 0.03$ | $\mathbf{2.93 \pm 0.02}$ | $3.06 \pm 0.18$ | $3.52 \pm 0.11$ | $3.19 \pm 0.05$ |
| Energy Efficiency | (768, 8) | $1.01 \pm 0.01$ | $1.21 \pm 0.01$ | $1.38 \pm 0.22$ | $\mathbf{0.78 \pm 0.06}$ | $1.01 \pm 0.04$ |
| Kin8nm | (8192, 8) | $\mathbf{-1.28 \pm 0.01}$ | $-1.14 \pm 0.01$ | $-1.20 \pm 0.02$ | $-1.11 \pm 0.01$ | $-1.15 \pm 0.01$ |
| Naval Propulsion | (11934, 16) | $-4.85 \pm 0.06$ | $-4.45 \pm 0.00$ | $-5.63 \pm 0.05$ | $\mathbf{-10.07 \pm 0.01}$ | $-10.01 \pm 0.01$ |
| Power Plant | (9568, 4) | $2.78 \pm 0.01$ | $2.80 \pm 0.01$ | $2.79 \pm 0.04$ | $2.94 \pm 0.01$ | $\mathbf{2.77 \pm 0.02}$ |
| Wine Quality Red | (1588, 11) | $0.97 \pm 0.01$ | $0.93 \pm 0.01$ | $0.94 \pm 0.12$ | $-0.78 \pm 0.07$ | $\mathbf{-0.98 \pm 0.06}$ |
| Yacht Hydrodynamics | (308, 6) | $1.64 \pm 0.02$ | $1.25 \pm 0.01$ | $1.18 \pm 0.21$ | $\mathbf{0.49 \pm 0.06}$ | $1.07 \pm 0.27$ |

where $(y_1, \ldots, y_d)$ is sampled from the posterior for $(x, y)$ and we used the independence of the posterior for each class.[8] Note that we also treat the predictions on different test points independently. In general, Eq. (6) does not have an analytic expression, and we resort to Monte-Carlo estimation. We refer readers to the supplement for comparison to other heuristics (*e.g.* passing the mean predictor through a softmax function and pairwise comparison). While this is heuristic, we find it is well calibrated (see Fig. 2). This is perhaps because the posterior Eq. (5) still represents substantial model averaging, and most uncertainty in high-SNR cases is epistemic rather than aleatory.

### 4.3 BENCHMARK ON CIFAR10

We examine the calibration of NNGP-R on increasingly corrupted images of CIFAR10-C (Hendrycks & Dietterich, 2019) using the benchmark of Ovadia et al. (2019). The results are displayed in Fig. 2. While FC-NNGP is similar to the standard RBF kernel, C-NNGP outperforms both in terms of calibration and accuracy. Moreover, we find that at severe corruption levels, C-NNGP actually outperforms all methods in Ovadia et al. (2019) (compare against their Table G1) in BS and ECE.

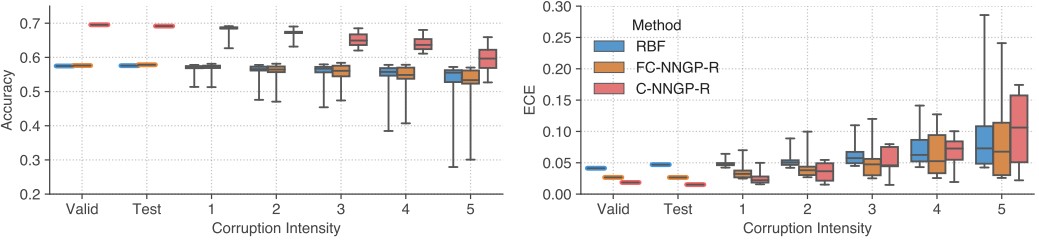

Figure 2: Uncertainty metrics across shift levels on CIFAR10 using NNGP-R. CNN kernels perform best as well as being more robust to corruption. See the supplement for numerical values at each quartile as well as a comparison to the NTK (Fig. S3). All methods remain well calibrated for all intensities of shifts, with C-NNGP-R performing best, and significantly better than methods in Ovadia et al. (2019); contrast against their Fig. 2 and S4.

---

[8]Implied directly by the NNGP's Kronecker structure and treating the labels as independent targets.

## 5  BAYESIAN OR INFINITE-WIDTH LAST LAYER

As we have seen NNGP-C and NNGP-R are remarkably well-calibrated. However, obtaining high performing models can be computationally intensive, especially for large datasets. NNGP-C and NNGP-R have running times that are cubic in the dataset size, due to computation of the kernel's Cholesky decomposition, with NNGP-C suffering additionally from potentially slow convergence of MCMC. Moreover, performant NNGP kernels require substantial compute to obtain (Novak et al., 2019b; Arora et al., 2019; Novak et al., 2019a; Li et al., 2019) in contrast to training a NN to similar accuracies. Moreover, even though the most performant NNGP kernels are SotA for a kernel method (Shankar et al., 2020), they still under-perform SotA NNs by a large margin. To combine the benefits of the NNGP and NNs, obtaining models that are both performant and well calibrated, we propose stacking an infinite-width sub-network on top of a pre-trained NN. More precisely, we use features obtained from a pre-trained model as inputs to the NNGP. As such, the outputs of the combined network are drawn from a GP and we may use Eqs. (5) and (6) for inference. We refer to this as neural network Gaussian process last-layer (NNGP-LL). Mathematically, the model is

$$y \sim f(g(x)) + \varepsilon, \text{ where } f \sim \mathcal{GP}(\mathbf{0}, \mathcal{K}), \tag{7}$$

where $g$ is a pre-trained embedding, $\mathcal{K}$ is the NNGP kernel, and $\varepsilon \sim \mathcal{N}(0, \sigma_\epsilon)$ is a noise term. For the implementation of this method, we imagine that $g$ is an input to the algorithm. The computation of the kernel requires precomputing the embeddings of all data points, which are then considered as inputs to the NNGP-R algorithm, resulting in a running time that is cubic in the training set size (see Sec. E). The main advantage is that the embedding $g$ allows for great expressive power which can be coupled with computationally cheaper kernels. This method draws inspiration from Hinton & Salakhutdinov (2008); Wilson et al. (2016b); Bradshaw et al. (2017), but with a *multi-layer* NNGP kernel. Note we are specifically interested in the calibration properties and so the innovations in computational efficiency in Wilson et al. (2016b) are complementary to our work.

This setup mirrors an important use case in practice, *e.g.* for customers of cloud ML services, who may use embeddings trained on vast amounts of non-domain-specific data, and then fine-tune this model to their specific use case. This fine tuning consists of either fitting a logisitic regression layer or deeper NNs using the embeddings obtained from their data, or perhaps training the whole NN generating the embedding by simply initializing with the pre-trained weights (Kornblith et al., 2019). These strategies allow practitioners to obtain highly accurate models without substantial data or computation. However, little is understood about the calibration of these transfer learning approaches.

We consider the EfficientNet-B3 (Tan & Le, 2019) embedding from TF-Hub[9] and TF Keras Applictions[10] that is trained on ImageNet (Deng et al., 2009), and perform our evaluations on CIFAR10 and its corruptions. We mainly use a multi-layer FC-NNGP as the top sub-network since its kernel is very fast to compute and the final FC layer of EfficientNet-B3 removes any spatial structure that might be exploited by convolutions. However, we also explore other kernel types (linear,[11] CNNs with pooling, and self-attention layers) in Fig. 3 by using earlier layers of EfficientNet-B3.

We compare our method with other popular last-layer methods for generating uncertainty estimates (Vanilla logisitic regression, using a deep NN for the last layers, temperature scaling (Platt et al., 1999; Guo et al., 2017), MC-Dropout (Gal & Ghahramani, 2016), ensembles of several last-layer deep NNs). In the supplement we further investigate these results with a WideResNet (Zagoruyko & Komodakis, 2016) that we can train from scratch, using the initialization method in Dauphin & Schoenholz (2019), which achieves good test performance on CIFAR-10 without BatchNorm (Ioffe & Szegedy, 2015). This allows us to compare against the gold standard ensemble method.

We find that in the transfer learning case, ensembles are quite ineffective alone. The best previous method is given by combining MC-Dropout, temperature scaling, and ensembles. However, this is still bested by NNGP-LL (see Table 3). We also examine the effect of the fine-tuning dataset size on calibration and performance. Remarkably, NNGP-LL is able to achieve accuracy and calibration comparable to ensembles with as few as 1000 training points.

---

[9] https://www.tensorflow.org/hub

[10] https://www.tensorflow.org/api_docs/python/tf/keras/applications

[11] In Riquelme et al. (2018), changing the last layer of a network to a Bayesian linear regression layer was useful for bandit tasks where calibrated uncertainty are important.

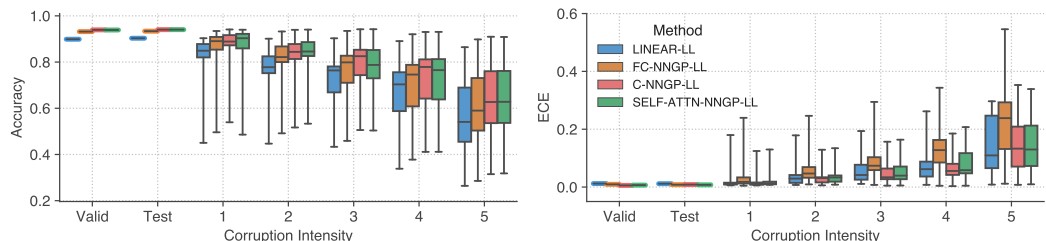

Figure 3: Uncertainty metrics across shift levels on CIFAR10 using NNGP-LL with EfficientNet-B3 embedding. We observe that more complex neural kernels (convolution and self-attention NNGPs) can have higher performance while being more robust to corruptions compared to replacing last layer to Bayesian linear regression (Riquelme et al., 2018). See Fig. S9 for the comparison to the NTK.

Table 3: NNGP-LL with EfficientNet-B3 fine tuned on CIFAR10 as an embedding and evaluated over all CIFAR10 corruptions and intensities. We show the quartiles over all corrupted variants for several different last-layer methods of obtaining uncertainties. *Vanilla* denotes fine tuned network, *Ensembles* refers to Lakshminarayanan et al. (2017) and *Ens/Drp/T* refers to combining Ensembles, MC-Dropout (Gal & Ghahramani, 2016) and temperature scaling (Guo et al., 2017). Center column is baseline where last-layer is replaced with an RBF kernel instead of NNGP. Right column groups are NNGP-LL results. More specifically, we train a three-layer FC network with dropout using input from the embedding from EfficientNet-B3 and report results in the middle columns. The rightmost columns show that NNGP-LL can be well-calibrated with very little training data (100, 1K, 5K, 10K, and full dataset are shown). See Fig. S5 for fine-grained box plots for each corruption level.

| Method | Vanilla | Ensembles | Ens/Drp/T | RBF-LL | 100 | 1K | 5K | 10K | NNGP-LL |
|---|---|---|---|---|---|---|---|---|---|
| Brier Score (25th) | 0.230 | 0.218 | 0.182 | 0.178 | 0.363 | 0.256 | 0.218 | 0.203 | **0.173** |
| Brier Score (50th) | 0.351 | 0.331 | **0.265** | 0.274 | 0.448 | 0.346 | 0.308 | 0.288 | 0.271 |
| Brier Score (75th) | 0.521 | 0.511 | 0.410 | **0.392** | 0.572 | 0.478 | 0.436 | 0.409 | 0.397 |
| NLL (25th) | 0.913 | 0.823 | 0.382 | 0.389 | 0.797 | 0.562 | 0.474 | 0.455 | **0.367** |
| NLL (50th) | 1.517 | 1.411 | **0.569** | 0.597 | 0.991 | 0.746 | 0.682 | 0.655 | 0.586 |
| NLL (75th) | 2.662 | 2.492 | 0.932 | **0.880** | 1.326 | 1.075 | 0.972 | 0.921 | 0.905 |
| ECE (25th) | 0.104 | 0.098 | **0.016** | 0.020 | 0.023 | 0.044 | 0.018 | 0.019 | 0.017 |
| ECE (50th) | 0.160 | 0.154 | 0.028 | 0.030 | 0.040 | 0.062 | **0.023** | 0.027 | 0.025 |
| ECE (75th) | 0.247 | 0.243 | 0.079 | 0.066 | 0.081 | 0.104 | **0.042** | 0.057 | 0.044 |
| Accuracy (75th) | 0.869 | 0.875 | 0.875 | 0.879 | 0.742 | 0.825 | 0.851 | 0.860 | **0.884** |
| Accuracy (50th) | 0.802 | 0.812 | **0.813** | 0.809 | 0.674 | 0.758 | 0.784 | 0.798 | **0.813** |
| Accuracy (25th) | 0.714 | 0.719 | 0.718 | 0.719 | 0.582 | 0.663 | 0.689 | 0.704 | **0.722** |

# 6 DISCUSSION

In this work, we explored several methods that exploit neural networks' implicit priors over functions in order to generate uncertainty estimates, using the corresponding Neural Network Gaussian Process (NNGP) as a means to harness the full distribution over function-space in the infinite-width limit. Using the NNGP, we performed fully Bayesian classification (NNGP-C) and regression (NNGP-R) and also examined heuristics for generating confidence estimates when classifying via regression. Across the board, we found that the NNGP provides good uncertainty estimates and generally delivers well-calibrated models even on OOD data. We found that NNGP-R is competitive with SOTA methods on the UCI regression task and remained calibrated even for severe levels of corruption. Despite their good calibration properties, as pure kernel methods, NNGP-C and NNGP-R cannot always compete with modern NNs in terms of accuracy. Adding an NNGP to the last-layer of a pre-trained model (NNGP-LL), allowed us to simultaneously obtain high accuracy and improved calibration. Moreover, we found NNGP-LL to be a simple and efficient way to generate uncertainty estimates with potentially very little data, and that it outperforms all other last-layer methods for generating uncertainties we studied. Overall, we believe that the infinite-width limit provides a promising direction to improve and better understand uncertainty estimates for NNs.

ACKNOWLEDGEMENT

We would like to thank Roman Novak, Jascha Sohl-Dickstein, and Florian Wenzel for insightful discussions and feedback.

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

# Supplementary Material

## A  DETAILED DESCRIPTION OF THE NNGP AND NTK

In this section, we describe the FC-NNGP and the C-NNGP. Most of the contents are adopted from Lee et al. (2018); Novak et al. (2019b); Lee et al. (2019), which we refer readers to for more technical details.

**NNGP:** Let $\mathcal{D} \subseteq \mathbb{R}^{n_0} \times \mathbb{R}^K$ denote the training set and $\mathcal{X} = \{x : (x, y) \in \mathcal{D}\}$ and $\mathcal{Y} = \{y : (x, y) \in \mathcal{D}\}$ denote the inputs and labels, respectively. Consider a fully-connected feed-forward network with $L$ hidden layers with widths $n_l$, for $l = 1, ..., L$ and a readout layer with $n_{L+1} = K$. For each $x \in \mathbb{R}^{n_0}$, we use $h^l(x), x^l(x) \in \mathbb{R}^{n_l}$ to represent the pre- and post-activation functions at layer $l$ with input $x$. The recurrence relation for a feed-forward network is defined as

$$\begin{cases} h^{l+1} &= x^l W^{l+1} + b^{l+1} \\ x^{l+1} &= \phi\left(h^{l+1}\right) \end{cases} \text{ and } \begin{cases} W_{i,j}^l &= \frac{\sigma_\omega}{\sqrt{n_l}}\omega_{ij}^l \\ b_j^l &= \sigma_b\beta_j^l \end{cases}, \tag{S1}$$

where $\phi$ is a point-wise activation function, $W^{l+1} \in \mathbb{R}^{n_l \times n_{l+1}}$ and $b^{l+1} \in \mathbb{R}^{n_{l+1}}$ are the weights and biases, $\omega_{ij}^l$ and $b_j^l$ are the trainable variables, drawn i.i.d. from a standard Gaussian $\omega_{ij}^l, \beta_j^l \sim \mathcal{N}(0, 1)$ at initialization, and $\sigma_\omega^2$ and $\sigma_b^2$ are weight and bias variances.

As the width of the hidden layers approaches infinity, the Central Limit Theorem (CLT) implies that the outputs at initialization $\{f(x)\}_{x \in \mathcal{X}}$ converge to a multivariate Gaussian in distribution. Informally, this occurs because the pre-activations at each layer are a sum of Gaussian random variables (the weights and bias), and thus become a Gaussian random variable themselves. See Poole et al. (2016); Schoenholz et al. (2017); Lee et al. (2018); Xiao et al. (2018); Yang & Schoenholz (2017) for more details, and Matthews et al. (2018a); Novak et al. (2019b) for a formal treatment.

Therefore, randomly initialized neural networks are in correspondence with a certain class of GPs (hereinafter referred to as NNGPs), which facilitates a fully Bayesian treatment of neural networks (Lee et al., 2018; Matthews et al., 2018b). More precisely, let $f^i$ denote the $i$-th output dimension and $\mathcal{K}$ denote the sample-to-sample kernel function (of the pre-activation) of the outputs in the infinite width setting,

$$\mathcal{K}^{i,j}(x, x') = \lim_{\min(n_1,...,n_L) \to \infty} \mathbb{E}\left[f^i(x) \cdot f^j(x')\right], \tag{S2}$$

then $f(\mathcal{X}) \sim \mathcal{N}(0, \mathcal{K}(\mathcal{X}, \mathcal{X}))$, where $\mathcal{K}^{i,j}(x, x')$ denotes the covariance between the $i$-th output of $x$ and $j$-th output of $x'$, which can be computed recursively (see Lee et al. (2018, §2.3). For a test input $x \in \mathcal{X}_T$, the joint output distribution $f([x, \mathcal{X}])$ is also multivariate Gaussian. Conditioning on the training samples, $f(\mathcal{X}) = \mathcal{Y}$, the distribution of $f(x)| \mathcal{X}, \mathcal{Y}$ is also a Gaussian $\mathcal{N}\left(\mu(x), \sigma^2(x)\right)$,

$$\mu(x) = \mathcal{K}(x, \mathcal{X})\mathcal{K}^{-1}\mathcal{Y}, \quad \sigma^2(x) = \mathcal{K}(x, x) - \mathcal{K}(x, \mathcal{X})\mathcal{K}^{-1}\mathcal{K}(x, \mathcal{X})^T, \tag{S3}$$

and where $\mathcal{K} = \mathcal{K}(\mathcal{X}, \mathcal{X})$. This is the posterior predictive distribution resulting from exact Bayesian inference in an infinitely-wide neural network.

**C-NNGP:** The above arguments can be extended to convolutional architectures Novak et al. (2019b). By taking the number of channels in the hidden layers to infinity simultaneously, the outputs of CNNs also converge weakly to a Gaussian process (C-NNGP). The kernel of the C-NNGP takes into account the correlation between pixels in different spatial locations and can also be computed exactly via a recursively formula; *e.g.*, see Eq. (7) in (Novak et al., 2019b, §2.2). Note that for convolutional architectures, there are two canonical ways of collapsing image-shaped data into logits. One is to vectorlize the image to a one-dimensional vector (CNN-Vec) and the other is to apply a global average pooling to the spatial dimensions (CNN-GAP). The kernels induced by these two approaches are very different and so are the C-NNGPs. We refer the readers to Section 3.2 of (Novak et al., 2019b, §3) for more details. In this paper, we have focused mostly on vectorization since it is more efficient to compute.

**ATTN-NNGP:** There is also a correspondence between self-attention mechanisms and GPs. Indeed, for multi-head attention architectures, as the number of heads and the number of features tend to infinity, the outputs of an attention model also converge to a GP (Hron et al., 2020). We refer the readers to Hron et al. (2020) for technical details.

**NTK:**  When neural networks are optimized using continuous gradient descent with learning rate $\eta$ on mean squared error (MSE) loss, the function evaluated on training points evolves as

$$\partial_t f_t(\mathcal{X}) = -\eta J_t(\mathcal{X}) J_t(\mathcal{X})^T \left( f_t(\mathcal{X}) - \mathcal{Y} \right) \tag{S4}$$

where $J_t(\mathcal{X})$ is the Jacobian of the output $f_t$ evaluated at $\mathcal{X}$ and $\Theta_t(\mathcal{X}, \mathcal{X}) = J_t(\mathcal{X}) J_t(\mathcal{X})^T$ is the Neural Tangent Kernel (NTK). In the infinite-width limit, the NTK remains constant ($\Theta_t = \Theta$) throughout training (Jacot et al., 2018). Thus the above equation is reduced to a constant coefficient ODE

$$\partial_t f_t(\mathcal{X}) = -\eta \Theta \left( f_t(\mathcal{X}) - \mathcal{Y} \right) \tag{S5}$$

and the time-evolution of the outputs of unseen input $\mathcal{X}_T$ can be solved in closed form as a Gaussian with mean and covariance

$$\mu(\mathcal{X}_T) = \Theta \left( \mathcal{X}_T, \mathcal{X} \right) \Theta^{-1} \left( I - e^{-\eta \Theta t} \right) \mathcal{Y}, \tag{S6}$$

$$\Sigma(\mathcal{X}_T, \mathcal{X}_T) = \mathcal{K} \left( \mathcal{X}_T, \mathcal{X}_T \right) + \Theta(\mathcal{X}_T, \mathcal{X}) \Theta^{-1} \left( I - e^{-\eta \Theta t} \right) \mathcal{K} \left( I - e^{-\eta \Theta t} \right) \Theta^{-1} \Theta \left( \mathcal{X}, \mathcal{X}_T \right)$$
$$- \left( \Theta(\mathcal{X}_T, \mathcal{X}) \Theta^{-1} \left( I - e^{-\eta \Theta t} \right) \mathcal{K} \left( \mathcal{X}, \mathcal{X}_T \right) + h.c. \right). \tag{S7}$$

Note that the randomness of the solution is the consequence of the random initialization $f(\mathcal{X}) \sim \mathcal{N}(0, \mathcal{K}(\mathcal{X}, \mathcal{X}))$.

Finally, the above arguments do not rely on the choice of architectures and we could likewise define CNTK and ATTN-NTK, the NTK for CNNs and attention models, respectively.

## B  ADDITIONAL FIGURES FOR NNGP-C

In this section, we show some additional plots and results comparing NNGP-C against standard NNs. Recall that in the NNGP-C model inference was carried out by elliptical slice sampling[12] (Murray et al., 2010), which is hyperparameter free, but there are hyperparameter associated with the NNGP and NNs. Here, we mainly address the method of hyperparameter tuning considered in the main text, where we fixed the hyperparameters that are common to both the NNGP and the NN, then only tuned the additional NN hyperparameters. We show results for tuning all of the NN's hyperparameter from scratch.

**Additional Tuning details.**    For any tuning of hyperparameters, we split the original training set of CIFAR10 into a 45K training set and a 5K validation set. All models were trained using the 45K points, and we then selected the hyperparameters from the validation set performance. We introduced a constant that multiples the whole NNGP kernel, or equivalently scales the whole latent space vector or the last layer bias and weight standard deviations—we called this constant the kernel scale. For FC-NNGP-C, the activation function was tuned over $\{\mathrm{ReLU}, \mathrm{erf}\}$, the weight standard deviation was tuned over $[0.1, 2.0]$ on a linear scale, the bias standard deviation was tuned over $[0.1, 0.5]$ on a linear scale, the kernel scale was tuned over $[10^{-2}, 100]$ on a logarithmic scale, the depth was tuned over $\{1, 2, 3, 4, 5\}$, and the diagonal regularizer was tuned over $[0., 0.01]$ on a linear scale.

For the FC-NN, there are additional hyperparameters: the learning rate, training steps, and width. For the NN, we considered two types of tuning. Either, as in the main text, the hyperparameters that are shared with the NNGP are fixed and the additional hyperparameters are tuned, or, as we present in the supplement, all of the NN's hyperparameters are tuned from scratch. In either case, the activation, the weight standard deviation, the bias standard deviation, the kernel scale, and the depth were tuned as above. The learning rate was tuned over $[10^{-4}, 0.1]$ on a logarithmic scale, the total training steps was tuned over $[10^5, 10^7]$ on a logarithmic, and the width was tuned over $\{64, 128, 256, 512\}$.

For the C-NNGP-C, all hypermaramters were treated as for the FC-NNGP-C case, except depth which was limited to $\{1, 2\}$. For the CNN, we again considered the two types of tuning: either fixing common hyperparameters or retuning all hyperparameters from scratch. The CNN's learning rate was tuned over $[10^{-4}, 0.1]$ on a logarithmic scale, the total training steps was tuned over $[2^{17}, 2^{22}]$ on a logarthimic scale, and the width was tuned over $\{64, 128, 256, 512\}$.

---

[12]Our implementation of ESS in JAX that is suitable for GPUs and TPUs can be found here: `https://github.com/google-research/google-research/tree/master/infinite_uncertainty`.

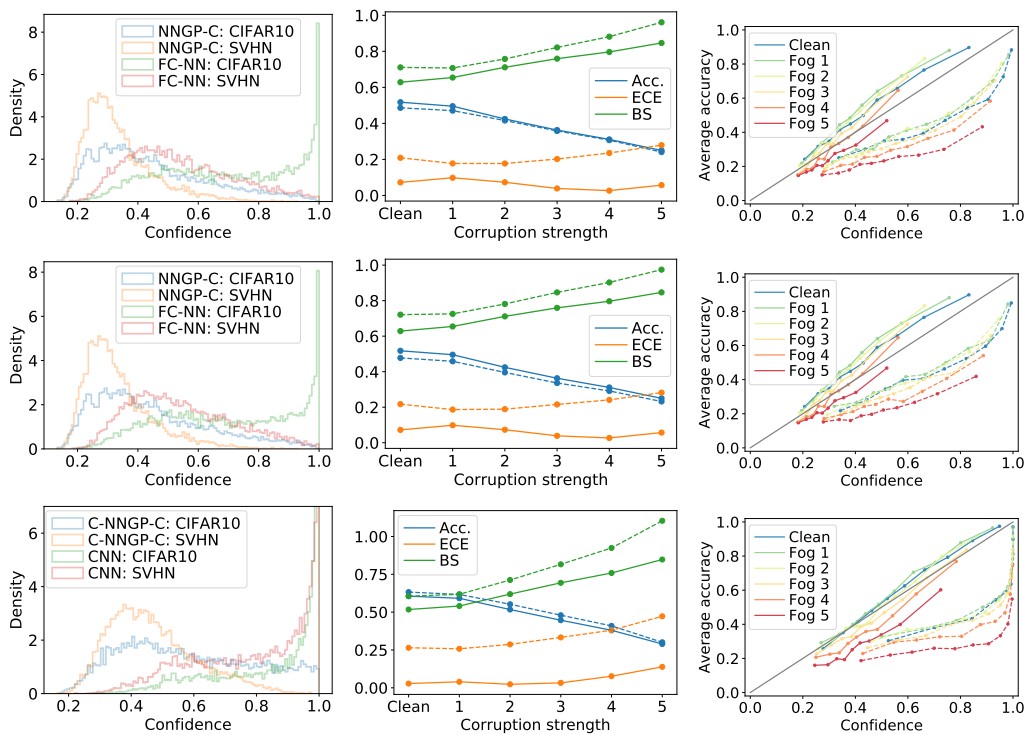

Figure S1: Investigating the calibration of Gaussian process classification with NNGP kernels as in Fig. 1, where we find similar results. (**left column**) Histogram of the confidence of the posterior distribution for each test point. We compare the NNGP-C and a finite width NN on an identically distributed test set (CIFAR10) and an OOD test set (SVHN). (**middle column**) Performance, NNGP-C is solid and NN dashed, under increasing distributional shift given by the CIFAR10 fog corruption. (**right column**) We bin the test set into ten bins sorted by confidence, and we plot mean confidence against mean accuracy. (**top row**) We compare FC-NNGP-C against a FC-NN with the same hyperparameters. (**middle row**) We compare FC-NNGP-C against a FC-NN, where all of the NN's hyperparameters are tuned independently with Vizier. (**bottom row**) We compare C-NNGP-C against a CNN, where all of the CNN's hyperparameters are tuned independently with Vizier.

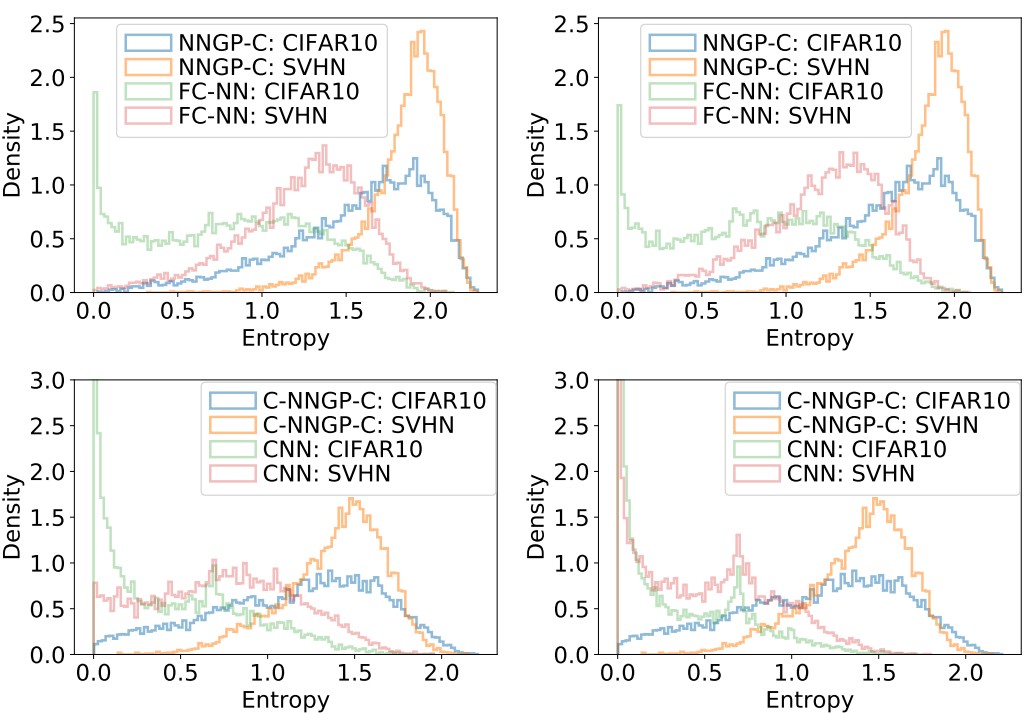

Figure S2: We compare NNGP-C and a finite width NN on an identically distributed test set (CIFAR10) and an OOD test set (SVHN). We plot a histogram of the entropy of the predicted distribution for each example in the test set. (**top row**) Fully connected network architectures. (**bottom row**) Convolutional architectures. (**left column**) Same hyperparamers. (**right column**) All of the CNN's hyperparameters are tuned independently.

Table S1: Performance of NNs, where all of the NN's hyperparameters are tuned independently with Google Vizier hyperparameter tuner (Golovin et al., 2017a). In addition we show results on all Fog corruptions that were omitted from Table 1.

| Data | Metric | FC | | | CNN | | |
|---|---|---|---|---|---|---|---|
| | | NN (same) | NN (all tuned) | NNGP-C | NN (same) | NN (all tuned) | NNGP-C |
| CIFAR10 | ECE | 0.209 | 0.217 | **0.072** | 0.283 | 0.265 | **0.031** |
| | Brier Score | 0.711 | 0.721 | **0.629** | 0.685 | 0.605 | **0.519** |
| | Accuracy | 0.486 | 0.478 | **0.518** | 0.576 | **0.632** | 0.609 |
| | NLL | 17383 | 17967 | **14007** | 21786 | 21160 | **11215** |
| Fog 1 | ECE | 0.178 | 0.187 | **0.098** | 0.271 | 0.258 | **0.042** |
| | Brier Score | 0.707 | 0.726 | **0.655** | 0.685 | 0.618 | **0.542** |
| | Accuracy | 0.471 | 0.458 | **0.496** | 0.561 | **0.617** | 0.590 |
| | NLL | 16702 | 17372 | **14671** | 19716 | 19484 | **11755** |
| Fog 2 | ECE | 0.177 | 0.189 | **0.073** | 0.286 | 0.287 | **0.018** |
| | Brier Score | 0.758 | 0.781 | **0.711** | 0.747 | 0.712 | **0.620** |
| | Accuracy | 0.416 | 0.396 | **0.425** | 0.507 | **0.552** | 0.519 |
| | NLL | 18026 | 18930 | **16233** | 20822 | 21657 | **13819** |
| Fog 3 | ECE | 0.202 | 0.215 | **0.039** | 0.320 | 0.333 | **0.03** |
| | Brier Score | 0.822 | 0.846 | **0.759** | 0.836 | 0.816 | **0.694** |
| | Accuracy | 0.358 | 0.336 | **0.363** | 0.445 | **0.480** | 0.445 |
| | NLL | 20092 | 21247 | **17638** | 23752 | 25703 | **16019** |
| Fog 4 | ECE | 0.236 | 0.241 | **0.026** | 0.352 | 0.381 | **0.071** |
| | Brier Score | 0.881 | 0.902 | **0.797** | 0.914 | 0.924 | **0.758** |
| | Accuracy | 0.307 | 0.292 | **0.311** | 0.385 | **0.409** | 0.380 |
| | NLL | 22589 | 23746 | **18867** | 27055 | 30844 | **18353** |
| Fog 5 | ECE | 0.279 | 0.282 | **0.057** | 0.420 | 0.472 | **0.134** |
| | Brier Score | 0.961 | 0.975 | **0.847** | 1.052 | 1.104 | **0.846** |
| | Accuracy | 0.241 | 0.232 | **0.250** | 0.287 | **0.301** | 0.289 |
| | NLL | 26345 | 27493 | **20694** | 33891 | 41058 | **22014** |
| SVHN | Conf. | 0.537 | 0.542 | 0.335 | 0.718 | 0.794 | 0.463 |
| | Entropy | 1.230 | 1.208 | 1.840 | 0.733 | 0.524 | 1.403 |
| CIFAR100 | Conf. | 0.651 | 0.654 | 0.398 | 0.812 | 0.847 | 0.474 |
| | Entropy | 0.944 | 0.930 | 1.663 | 0.493 | 0.394 | 1.420 |

## C  COMPARISON OF HEURISTICS FOR GENERATING CONFIDENCES FROM NNGP-R

In Secs. 4 and 5, we utilized a heuristic to generate confidence from exact GPR posterior distribution. Here we denote the heuristic described in Eq. (6) as *exact* confidence, which is the probability of a class probit being maximal under an independent multivariate Gaussian distribution. We consider two more heuristics. One is denoted *pairwise*, where we take confidence to be proportional to the probability that the $i$-th class probit is larger than other probits in pairwise fashion, *i.e.*

$$p_x(i) \propto \mathbb{P}[z_i > z_j, \forall j \neq i] = \prod_{j \neq i} p(y_i > y_j | x, \mathcal{X}, \mathcal{Y}) = \prod_{j \neq i} \Phi \left( \frac{\mu_i - \mu_j}{\sqrt{\sigma_i^2 + \sigma_j^2}} \right) , \qquad \text{(S8)}$$

where $\Phi(\cdot)$ is Gaussian cumulative distribution function. In order to obtain confidence, we normalize by the sum so that the heuristic confidence sums up to 1. This is following the one-vs-one multiclass classification strategy Hastie & Tibshirani (1998).

We note that, we introduce temperature scaling with temperature $T$ by replacing posterior variances as

$$\sigma_T^2 = T\sigma^2 . \qquad \text{(S9)}$$

Another heuristic is denoted *softmax*, where we apply the softmax function to the posterior mean:

$$p_x(i) := \sigma(\mu)_i = \frac{e^{\mu_i/\sqrt{T}}}{\sum_j e^{\mu_j/\sqrt{T}}} . \qquad \text{(S10)}$$

In this case, the posterior variance is not used to construct the heuristic confidences.

A comparison for these three-different heuristics for C-NNGP-R is shown in Fig. S4 with and without temperature scaling. We note that *exact* and *pairwise* heuristics remain well calibrated without temperature scaling. However with temperature scaling the *softmax* heuristic can be competitive to other heuristics. In Sec. 4 and 5, we focused on the *exact* heuristic.

Table S2: Quartiles of Brier score, negative-log-likelihood and ECE over all CIFAR10 corruptions for methods in Fig. 2.

| Method/Metric | RBF | FC-NNGP-R | C-NNGP-R | Myrtle-NNGP-R |
|---|---|---|---|---|
| Brier Score (25th) | 0.568 | 0.569 | 0.435 | **0.330** |
| Brier Score (50th) | 0.580 | 0.586 | 0.464 | **0.460** |
| Brier Score (75th) | 0.599 | 0.613 | **0.515** | 0.760 |
| Gaussian NLL (25th) | 0.147 | 0.612 | **0.108** | 0.763 |
| Gaussian NLL (50th) | **0.270** | 0.830 | 0.457 | 2.740 |
| Gaussian NLL (75th) | **0.447** | 1.099 | 1.027 | 6.933 |
| NLL (25th) | 1.331 | 1.351 | 1.002 | **0.783** |
| NLL (50th) | 1.363 | 1.398 | 1.079 | **1.050** |
| NLL (75th) | 1.415 | 1.466 | **1.178** | 1.773 |
| ECE (25th) | 0.048 | 0.030 | **0.022** | 0.035 |
| ECE (50th) | 0.052 | **0.039** | 0.046 | 0.085 |
| ECE (75th) | 0.069 | **0.065** | 0.071 | 0.203 |
| Accuracy (75th) | 0.573 | 0.574 | 0.683 | **0.770** |
| Accuracy (50th) | 0.566 | 0.561 | 0.659 | **0.678** |
| Accuracy (25th) | 0.549 | 0.541 | **0.628** | 0.432 |

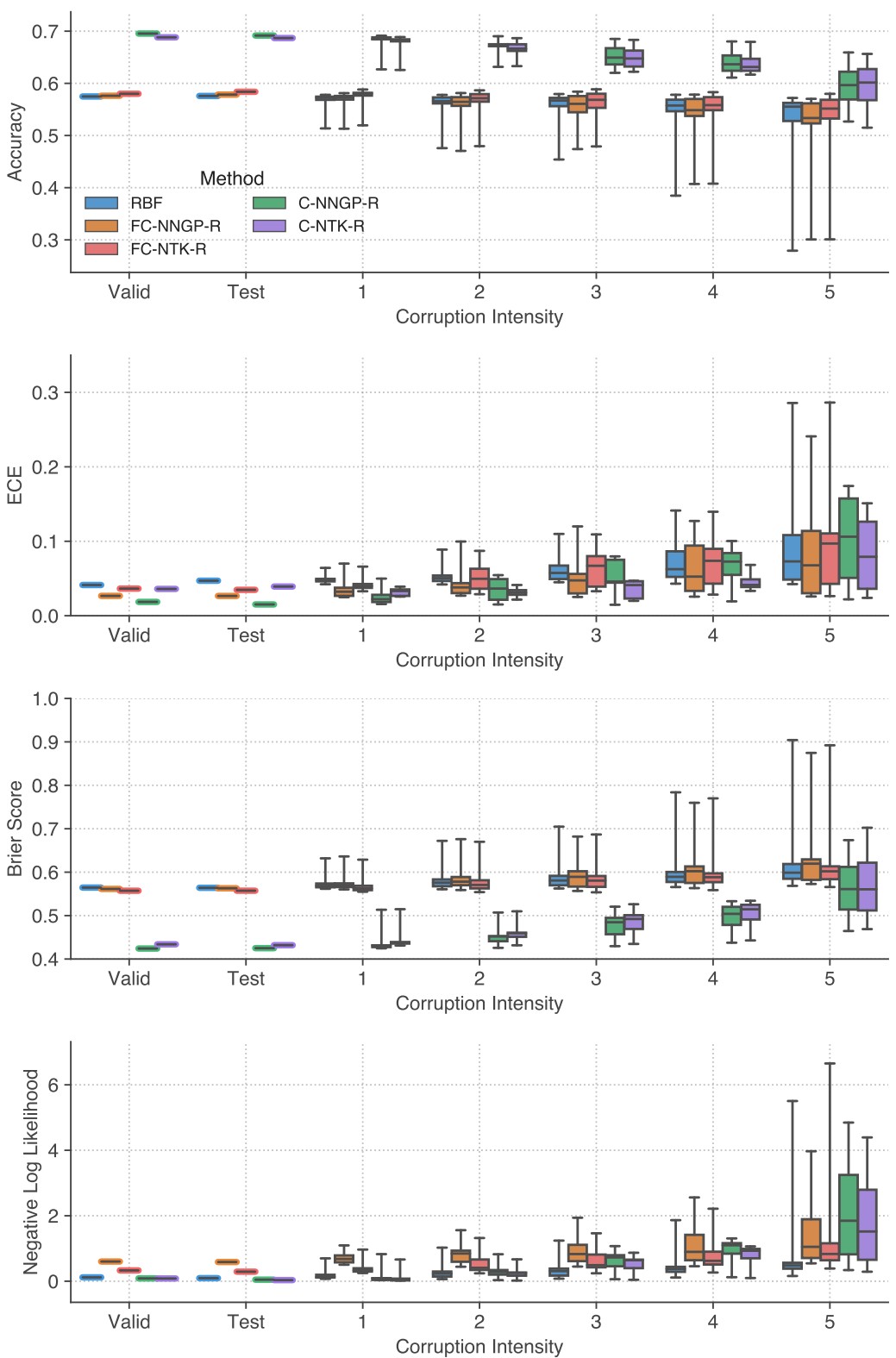

Figure S3: Uncertainty metrics across shift levels on CIFAR10 using NNGP-R and NTK-R. CNN kernels perform best as well as being more robust to corruption. The NTK corresponding to the same architecture shows similar robustness properties.

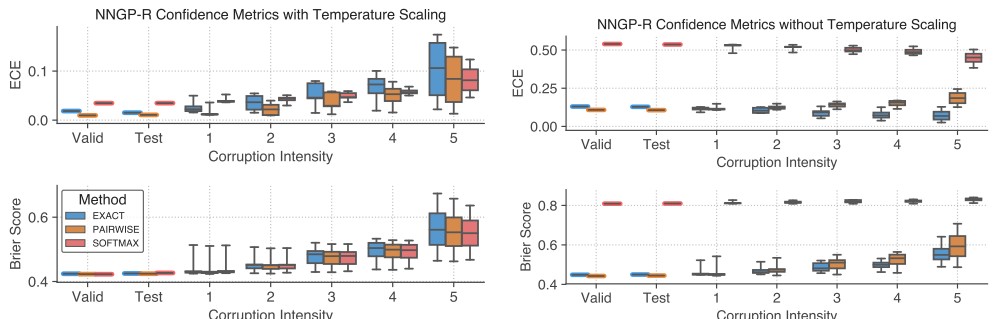

Figure S4: Comparison of NNGP-R based confidence measures on CIFAR10 corruptions using C-NNGP-R. Left column uses temperature scaling based on a validation set whereas right column uses $T = 1$. We see that softmax confidence requires adjusting temperature which is equivalent to modifying prior variance to be calibrated whereas exact and pairwise heuristic confidence is calibrated without needing to modify the prior variance.

# D  ADDITIONAL FIGURES FOR NNGP-LL

The results in the main text focused on a fixed embedding (see Table 3) and we show additional results for this as a box-plot in Fig. S5 here. However, it is also common in practice to tune all of the embedding's weights and simply initialize at their pre-triained values. We explore this setting in the supplement in Table S3 and Fig. S6 by considering the EfficientNet-B3 embedding and fine tuning it on CIFAR10. We also show a results comparing the FC-NNGP-LL with using the standard RBF kernel for the same purpose (see Fig. S7).

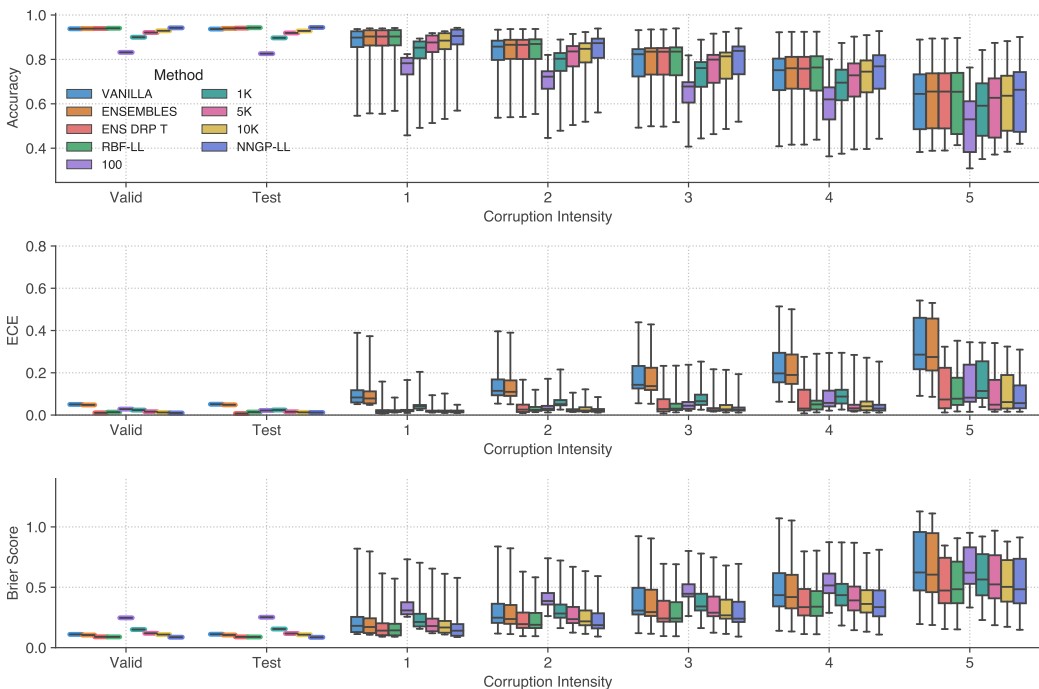

Figure S5: Uncertainty metrics across corruption levels on CIFAR10 using NNGP-LL with EfficientNet-B3 embedding. Baseline NNs are trained on CIFAR10 with parameters of body network fixed. See Table 3 for quartile comparison and Fig. S6 and Table S3 for comparison with fine tuning of all the embedding's weights. Fig. S7 compares use of NNGP and RBF kernel for NNGP-LL settings.

## D.1  A COMPARISON OF ENSEMBLE AND NNGP-LL ON WIDERESNET

To compare the NNGP-LL method against the gold standard ensemble method, we train a WideResnet 28-10 on CIFAR-10 from scratch with 5 different random initialization. The model is trained using MetaInit (Dauphin & Schoenholz, 2019), Delta-Orthogonal (Xiao et al., 2018), mixup (Zhang et al., 2017) and without BatchNorm (Ioffe & Szegedy, 2015). The model achieves about $94\%$ accuracy on the clean test set. See Table S4 and Fig. S8 for the comparison. We find the ECE for NNGP-LL is very competitive, even with small dataset size, compared to baseline methods including ensembles.

# E  COMPLEXITY OF NNGP-LL

There are two main steps in applying NNGP-LL. One is computing the NNGP kernel $\mathcal{K}$ and the other is inference. For NNGP-LL, we note that the computational of the kernel has the added cost of inference using the pretrained NN on the whole dataset, which adds a cost that is linear in dataset size but with a potentially large constant factor. For inference, we need to invert the whole kernel matrix of size $c|\mathcal{X}| \times c|\mathcal{X}|$, the computational complexity is usually $O(c^3|\mathcal{X}|^3)$ where $c$ is the number of classes and $|\mathcal{X}|$ is the training set size. However, since $\mathcal{K}$ has a Kronecker factorization, the complexity is reduced to $O(|\mathcal{X}|^3)$.

Table S3: Quartiles of Brier score, negative-log-likelihood, and ECE over all CIFAR10 corruptions for methods in Fig. S6 using EfficientNet-B3 embedding. The first five columns are the same as the methods in Table 3, whereas the last two columns use the same embedding architecture but where all the weights are fine tuned (FT) on CIFAR10. See Fig. S6 for the corresponding box-plot.

| | LL | LL T Scaling | LL Ens | LL Ens Drp T | NNGP-LL | FT-NN | FT-NNGP-LL |
|---|---|---|---|---|---|---|---|
| Brier Score (25th) | 0.230 | 0.191 | 0.218 | 0.182 | 0.173 | 0.083 | **0.081** |
| Brier Score (50th) | 0.351 | 0.281 | 0.331 | 0.265 | 0.271 | 0.155 | **0.153** |
| Brier Score (75th) | 0.521 | 0.422 | 0.511 | 0.410 | 0.397 | **0.341** | 0.361 |
| NLL (25th) | 0.913 | 0.406 | 0.823 | 0.382 | 0.367 | **0.166** | 0.196 |
| NLL (50th) | 1.517 | 0.612 | 1.411 | 0.569 | 0.586 | **0.320** | 0.374 |
| NLL (75th) | 2.662 | 0.988 | 2.492 | 0.932 | 0.905 | **0.730** | 0.868 |
| ECE (25th) | 0.104 | 0.016 | 0.098 | 0.016 | 0.017 | **0.005** | 0.012 |
| ECE (50th) | 0.160 | 0.030 | 0.154 | 0.028 | 0.025 | **0.015** | 0.022 |
| ECE (75th) | 0.247 | 0.092 | 0.243 | 0.079 | 0.044 | 0.051 | **0.046** |
| Accuracy (75th) | 0.869 | 0.869 | 0.875 | 0.875 | 0.884 | 0.945 | **0.947** |
| Accuracy (50th) | 0.802 | 0.802 | 0.812 | 0.813 | 0.813 | 0.894 | **0.896** |
| Accuracy (25th) | 0.714 | 0.714 | 0.719 | 0.718 | 0.722 | **0.758** | 0.748 |

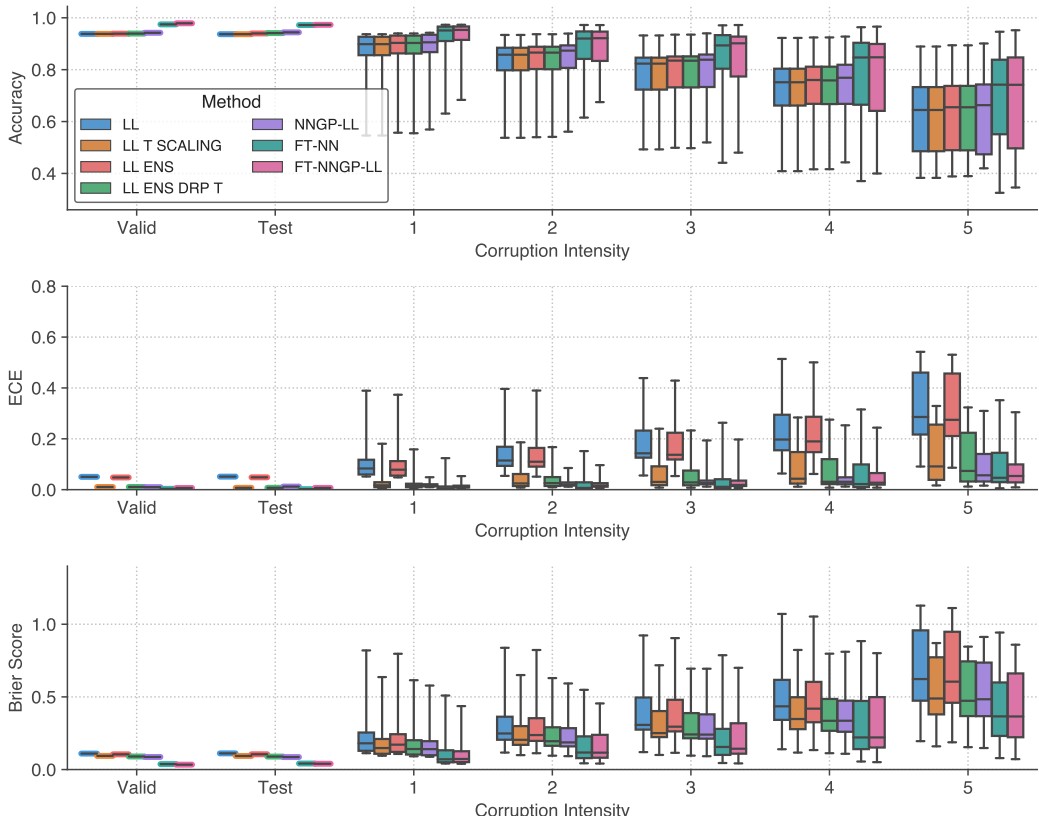

Figure S6: Uncertainty metrics across corruption levels on CIFAR10 using NNGP-LL with EfficientNet-B3 embedding. Baseline NNs are either last layer (LL) trained on CIFAR10 with parameters for body networks fixed or where all the weights are fine tuned (FT). Quartile comparisons can be found in Table S3.

The complexity of computing the NNGP kernels is architecture dependent. For notational simplicity, assume the cardinality of the training and test sets are $m$ and $n$.

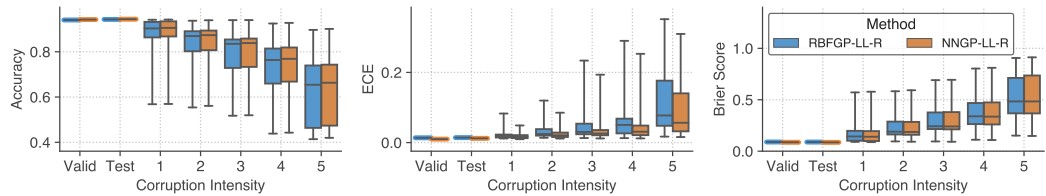

Figure S7: Comparison of NNGP-LL using NNGP head vs RBF GP head on EfficientNet-B3 embedding. While there are slight advantage of using NNGP head over RBF-GP overall they provide very similar benefits as corruption intensity increase.

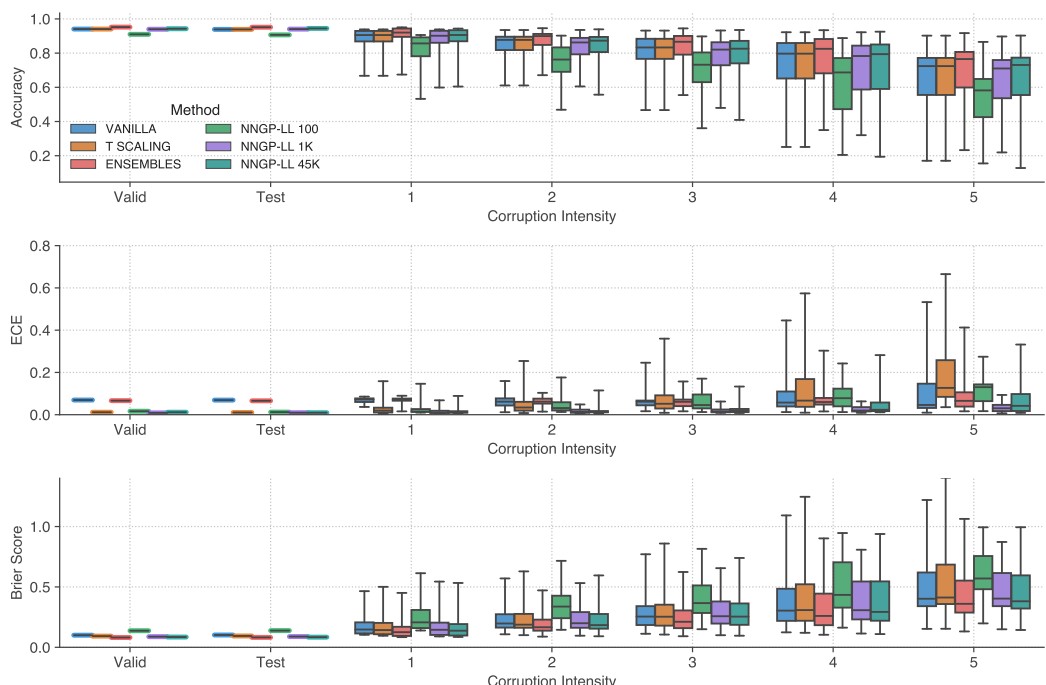

Figure S8: Uncertainty metrics across corruption levels on CIFAR10 using NNGP-LL with MetaInit embedding. Baseline NNs are compared with vanilla training, temperature scaling and ensembles.

### E.1 FCN

The FCN kernel is relatively cheap to compute since it depends only on the dot product between inputs. It takes $O(L(mn + n^2))$ time, where $L$ is the number of layers of the FCN. In practice, the whole CIFAR-10 kernel could be obtained within a minute.

### E.2 CNN-VEC

The CNN-VEC kernel is slightly more expensive to compute. Each entry $\mathcal{K}(x, x')$ is a function of $\{\langle x_a, x'_a \rangle : a \in [d]\}$ where $d$ is the total number of pixels in one image (e.g. $d = 32 * 32$ for CIFAR-10). Computing $\mathcal{K}(x, x')$ takes $O(Ld)$ time and the whole kernel takes $O((mn + n^2)Ld)$ time. In practice, the whole CIFAR-10 kernel could be obtained in about 80 minutes.

### E.3 CNN-GAP AND ATTENTION

CNN-GAP and Attention kernels are very expensive to compute. Each entry $\mathcal{K}(x, x')$ is a function of $\{\langle x_a, x'_b \rangle : a, b \in [d]\}$. The complexity for one entry is $O(d^2 L)$ and $O((mn + n^2)d^2 L)$ for the whole kernels. For a 10 layers Myrtle kernel, it takes about 400 GPU hours (using NVIDIA Tesla V100) to compute the upper triangle part for the whole CIFAR10 (where $d = 32^2$). However, for the last layer

Table S4: Quartiles of Brier score, negative-log-likelihood, ECE and accuray over all CIFAR10 corruptions for MetaInit Embedding NNGP-LL and MetaInit trained networks in Fig. S8.

| Method/Metric | Vanilla | T Scaling | Ensembles | 100 | 1K | NNGP-LL |
|---|---|---|---|---|---|---|
| Brier Score (25th) | 0.165 | 0.164 | **0.141** | 0.256 | 0.162 | 0.152 |
| Brier Score (50th) | 0.247 | 0.244 | **0.210** | 0.366 | 0.257 | 0.241 |
| Brier Score (75th) | 0.397 | 0.410 | **0.356** | 0.567 | 0.415 | 0.399 |
| NLL (25th) | 0.382 | 0.391 | **0.328** | 0.562 | 0.381 | 0.360 |
| NLL (50th) | 0.553 | 0.576 | **0.483** | 0.799 | 0.599 | 0.570 |
| NLL (75th) | 0.864 | 1.040 | **0.781** | 1.214 | 0.967 | 0.925 |
| ECE (25th) | 0.046 | 0.025 | 0.044 | 0.018 | **0.011** | 0.012 |
| ECE (50th) | 0.062 | 0.051 | 0.066 | 0.044 | **0.017** | **0.017** |
| ECE (75th) | 0.079 | 0.126 | 0.077 | 0.101 | **0.030** | 0.039 |
| Accuracy (75th) | 0.895 | 0.895 | **0.916** | 0.824 | 0.889 | 0.896 |
| Accuracy (50th) | 0.840 | 0.840 | **0.866** | 0.738 | 0.821 | 0.834 |
| Accuracy (25th) | 0.726 | 0.726 | **0.761** | 0.586 | 0.703 | 0.716 |

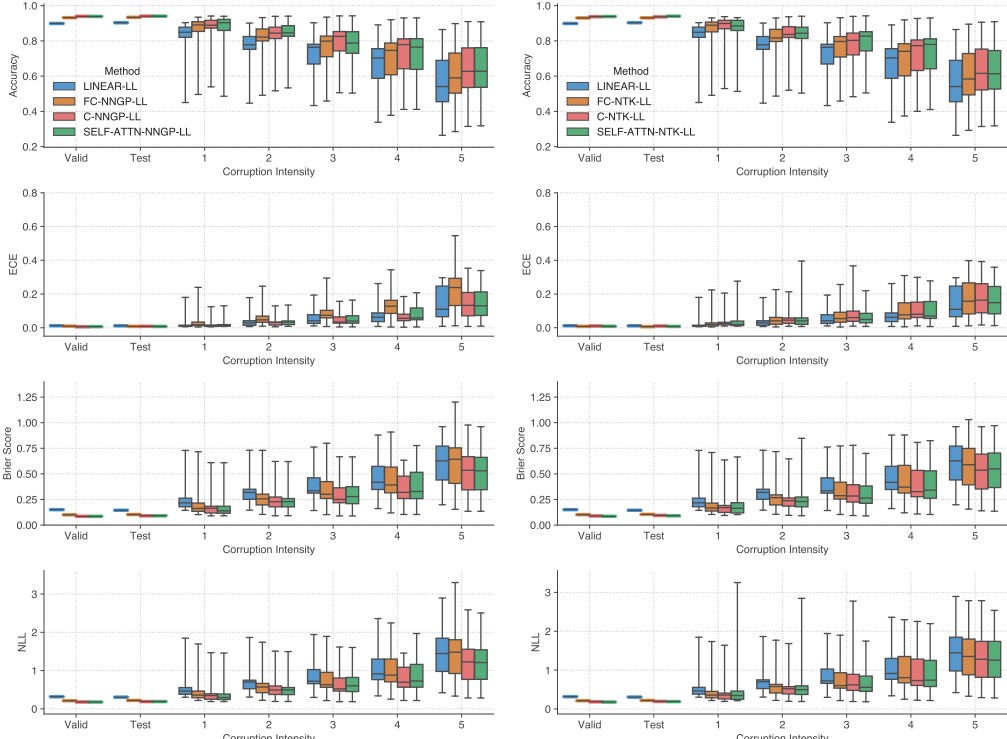

Figure S9: Comparison between NNGP-LL and NTK-LL with varying embeddings and neural kernels.

method, the images are down-sampled to smaller $8 \times 8$ image-patches. The whole kernel could be obtained in about $400/(4^4) \sim 1.6$ GPU hours.

## F    COMPARISON WITH MYRTLE ARCHITECTURE

While FCN or CNN-Vec NNGP performance is not as competitive as modern CNNs on CIFAR10 classification tasks, recently  Shankar et al. (2020) has shown that the Myrtle architecture along with ZCA regularized preprocessing can obtain highly performant kernels. Lee et al. (2020) has shown that this is generally true for pooling kernels.

Here we evaluate uncertainty properties of Myrtle-10 NNGP. Note that this kernel is extremely compute intensive (see SM E) and evaluation on the full CIFAR10 and CIFAR10-C requires about 12,000 GPU hours (using NVIDIA Tesla V100s). We release these precomputed kernels for any further analysis by the research community.[13]

We observe that NNGP-R with the Myrtle-10 kernel is the least robust under corruption when compared to other kernel architectures (Fig. S10). However, comparing to the finite Myrtle network trained via gradient descent, we observe that NNGP-R method is as robust as the deep ensemble method when average over 10 random seeds (Fig. S11). While we consider these results preliminary and requiring further investigation, it suggests there may be a necessary tradeoff between the accuracy and calibration obtainable by NNGP-R.

We also plot the singular values of the training kernels (10k training samples) for three different architectures: FCN, CNN-VEC and Myrtle (Fig. S12). Note that the Myrtle kernel is much more ill-conditioned than the other two kernels.

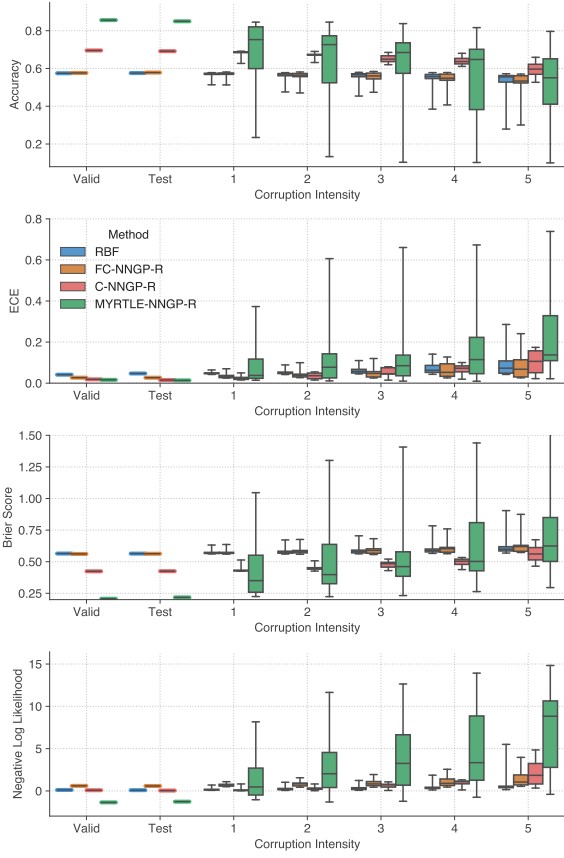

Figure S10: Uncertainty metrics across shift levels on CIFAR10 using NNGP-R compared with Myrtle-10 architecture (Shankar et al., 2020). Here we compare NNGP-R in Fig. 2 to the pooling based Myrtle-10 kernel with ReLU activation. All methods except for the Myrtle architecture remain well calibrated for all intensities of shifts. See Table S2 for detailed numbers.

---

[13]https://github.com/google-research/google-research/tree/master/infinite_uncertainty

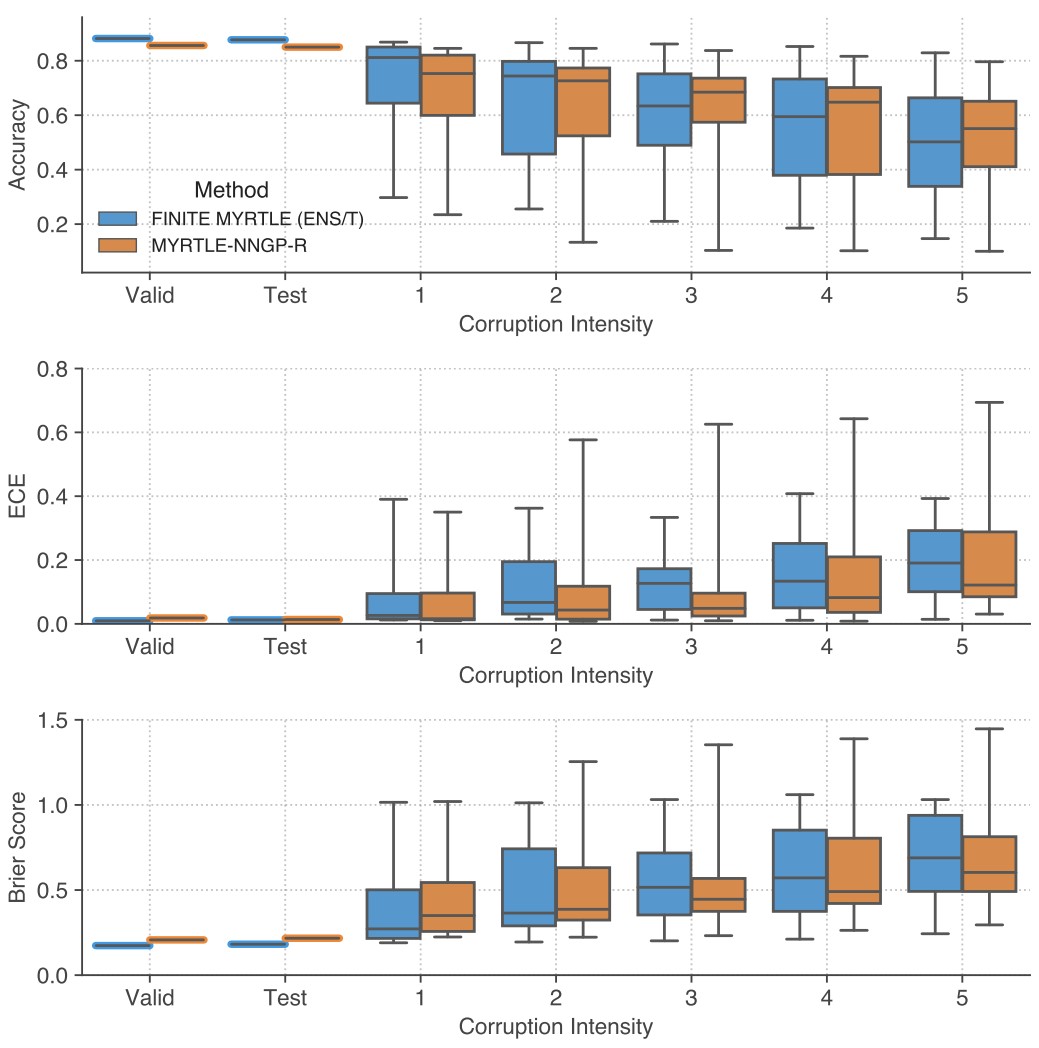

Figure S11: Uncertainty metrics across shift levels on CIFAR10 comparing Myrtle-NNGP-R and a deep ensemble of finite Myrtle-10 networks.

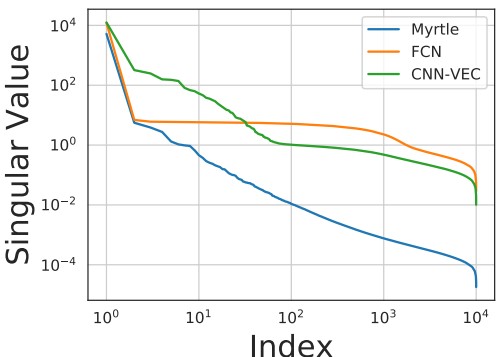

Figure S12: Singular values of the NNGP kernels of a 10k training set for three architectures: 3-layer fully-connected network (FCN), 10-layer convolutional network with a vectorization readout layer (CNN-VEC) and a 10-layer myrtle network (Myrtle).

