# OpenReview forum: "Exploring the Uncertainty Properties of Neural Networks’ Implicit Priors in the Infinite-Width Limit"
_ICLR.cc/2021/Conference — ICLR 2021 Poster_

### Official Review · AnonReviewer4 · 2020-10-28
**Good work, but overstated claims**

**Rating:** 6
**Confidence:** 4

**Review:**

After rebuttal: I am uneasy about the overstated claims made in section 2. That the architectures are small should really be mentioned more prominently. However reviewers #1 and #2 make a good case that what matters are the improvements presented in Section 5. Thus, I reluctantly recommend acceptance.

# Paper summary

The authors empirically investigate the calibration performance of NN-GPs in CIFAR10 and several UCI data sets, in three forms:
- Bayesian inference for the NN-GP function-space prior, through a softmax link function
- Heuristics to convert the classification-as-regression posterior into class probabilities
- As a "head" to a pre-trained network

Generally, they find the NN-GPs are competitive with the _examined_ alternatives.

# High-level comments

I think empirical work like this paper is important: we Bayesians like to justify ourselves using calibration, but unless the beautiful Bayesian methods are *actually* calibrated, they are not useful. This paper mainly explores some properties of existing algorithms that are not understood, which is great.

## Table 1's architectures are too small (2 layer CNN?)

However, I will criticise the methodology. The architecture of the NNs used in Table 1 is specified in the supplement, and it is not pretty: the CNN is restricted to have only 1 or 2 layers (Appendix B)q. The convolution filter shape is unspecified, so I'll assume it is 3x3 like most modern CNNs. Because these networks are so shallow, the receptive field of a given convolutional location is not even close to the full input image.

This architecture is simpler than even the 5-layer 1998 LeNet , and is very far from the 2015 ResNet (He et al.) that is a reasonable CIFAR10 baseline; let alone the current state of the art. It is also much simpler than the networks considered for kernel architectures by Shankar et al. (2020).

Given how far these architectures are from things used in practice that have decent CIFAR10 performance, I'm not sure we can extrapolate much from the data in Table 1. Perhaps the good calibration and superior performance of the kernel will stop being true once we go to 18 or 20 layers, or once we add mean-pooling to the CNN.  We know that the superior performance of the kernel stops being true at a certain point for classification-as-regression, and I see no reason to think it would be different for "proper" Bayesian inference. (In fact, in my experience, variational inference for GP classification with these kernels has worse accuracy than classification-as-regression).

It is true that calculating the kernel matrix for more complicated CNNs is much more expensive. However, you could do that experiment without tuning the hyperparameters with Vizier. Since the complexity of MCMC once you have the kernel matrix is the same, you should be able to do it. I suspect the accuracy will be much better for the modern CNNs without tuning, than for the 2-layer CNNs you found (the calibration may be better or worse).

At the very least, the caveat that the CNNs are tiny should be clearly stated in Section 3 (and not hidden in the appendix).

## Ignores similar work doing Bayesian linear regression on top of a NN base

The paper "Deep Bayesian bandits showdown" (https://arxiv.org/abs/1802.09127) compares various Bayesian deep learning methods on a bandit task, where calibrated uncertainty is crucial. As a result, most of them fail. The one that seems to come out on top is to change the last layer of a network to a Bayesian linear regression layer. This is equivalent to the method shown in Section 5 of this paper (except this paper sometimes uses a kernel, that is not just the linear regression kernel). It should be at least acknowledged, and ideally it would be included in Table 3.

Table 3 should also acknowledge Bradshaw et al. (2017, https://arxiv.org/abs/1707.02476), which adds an RBF kernel on top of a neural network. All of these should perform similarly to NNGP-LL, especially given how close the RBF kernel is to the NNGP in Table 2.

## Unclear that NNGPs have better uncertainty in Table 2

And that's okay! But it should be a bit more emphasized than "competitive in NLL" just before Section 4.2. Also, the non-standard training for RBF kernels (selecting hyperparameters on a validation set, instead of maximising the train likelihood) may hurt their performance in Table 2.

That NN-GPs are good in accuracy at UCI data sets was noted by Arora et al. (https://openreview.net/forum?id=rkl8sJBYvH), though only for classification

## In conclusion: overstated claims

I think this paper is good, but it overstates how good the methods evaluated turn out to be. It paints a rosy picture of everything, like calibration in NNs is now solved, but that is not true. However, I do think this paper is a meaningful contribution, that puts together a few ideas that were lying around and shows that you can use them to make progress in calibration in NNs.

The authors should clearly caveat in section 3 that the results may not extrapolate to bigger CNNs or, ideally, add a data point with bigger CNNs. The authors should compare to similar algorithms that will be competitive in Table 3.

# Minor points

Top of pg. 2: SNR -> show acronym.

before section 1.1: "It is possible to disambiguate between the uncertainty properties of the NN prior and those due to the specific optimization decisions by performing Bayesian inference". What do you mean? Transforming the prior into a posterior using Bayesian updating is one such decision. Also, because you're approximating the prior, deciding which Bayesian inference algorithm to use matters.

Section 3: "by avoiding heuristic approaches to inference, we are able to directly evaluate the prior". I realise now that by this you mean the same as what I asked above, whatever it is. But the phrasing would be clearer with "we are able to directly evaluate the [properties conferred by the] prior". The fact is, you can evaluate the prior density for any weight setting fairly trivially!

Page 3, top: "under gradient flow to minimize a loss, the output distribution remains a GP". This is only true for squared loss, as far as I know.

---

> ### Author Response · Authors · 2020-11-21
> **Response to Reviewer #4 (part 1)**
>
> We thank the reviewer for the effort in reading our paper and allowing us to discuss and clarify some of the weaknesses in our study.
>
>
> We agree that the CNNs used in Sec. 3 are far from state-of-the-art. The goal of this section was to make a direct comparison between the CNN-GPs and finite CNNs. Since the NNGP-C method is quite impractical computationally, we did not compare to the stronger baselines from Ovadia et al. (2019), unlike in other sections.
>
>
> Unfortunately, we found that the convergence of the elliptical slice sampling was highly sensitive to the kernel hyperparameters---much more so than for standard GP regression with the same kernel. This meant that tuning of the kernel hyperparameters was essential to get reasonable results. This also held true for the more performant, and more computationally expensive, Myrtle and global-average-pooling kernels. Since it is not feasible to re-compute these kernels many times to tune hyperparameters, we had to exclude them from this section of the paper. We have made this clearer in Sec. 3.
>
>
> Without tuning we were able to get around 69% accuracy on CIFAR10 with the Myrtle kernel with ESS using only 10,000 training points, but with the same hyperparameters and the full 45,000 training points, the ESS algorithm produced essentially random predictions. However, we did obtain results for the Myrtle kernel using the NNGP-R methodology. We have added a new section to the SM (SM Sec. F), but we consider these results preliminary and leave an in-depth analysis for future work. The most interesting finding is that the calibration of NNGP-R with the Myrtle kernel is similar to an ensemble of finite width networks of the same architecture. Consequently, the Myrtle kernel shows some of the same failings under distributional shift suggesting there may be a necessary tradeoff between accuracy and calibration that NNGP-R can be used to study.
>
>
> **[Bayesian Linear Regression]**: The Bayesian linear regression (studied by Riquelme et al., 2018) is a very intuitive baseline and we are happy to make comparisons to it. Originally, we did not include it in the submission because it is much weaker than other gold standard baselines e.g. deep ensembles (see table below).
>
> Though our simple FC NNGP-LL (i.e. using infinite-width FCN) is similar to classical deep kernel methods (e.g. RBF, Laplacian kernel), our main innovation here is the usage of the **spatially-aware** kernels: CNN with average pooling and self-attention architectures. Unlike classical kernels that are applied to the **vectorized** features (which is permutation invariant), our CNN-GAP/attention kernels are applied to the **un-vectorized** features and able to explore their spatial structure. As summarized in the table below (also Fig. 3), these spatially-aware kernels outperform classical kernels and finite width network ensembles in both accuracy and calibration.
>
> Note that linear regression is a very weak baseline, while more powerful kernels such as Attention-GP/ CNN-GAP-GP are much more expressive, better results than deep kernel / FC-NNGP kernel (see updated figure 3). Furthermore, note that we compare our methods to the current SotA baseline, deep ensembles (Lakshminarayanan et al. 2016), see figure 3.
>
> |                                             | Ensemble      | linear GP     | FC-NNGP        | C-NNGP         | Attention-GP   |
> |---------------------------------------------|---------------|---------------|----------------|----------------|----------------|
> | Accuracy  (Clean / top 50% all corruptions) | 94.0 / 81.2   | 90.3 / 76.3   | 93.4 / 80.2    | 94.0 / 82.4    | 94.0 / 80.8    |
> | ECE (Clean / top 50% all corruptions)       | 0.048 / 0.154 | 0.012 / 0.041 | 0.0082 / 0.070 | 0.0085 / 0.033 | 0.0077 / 0.040 |
> | Brier (Clean / top 50% all corruptions)     | 0.105 / 0.331 | 0.145 / 0.337 | 0.103/ 0.293   |  0.091 / 0.250 | 0.091 / 0.258  |

---

> > ### Author Response · Authors · 2020-11-21
> > **Response to Reviewer #4 (part 2)**
> >
> > Continued from (part 1)
> >
> > **[UCI Datasets]**: While we agree hyperparameter tuning with cross-validation is not a standard procedure for Bayesian models, Mukhoti et al., (2018), for example, showed that this can lead to significantly better performance on RMSE and NLL for the Dropout method. While we have not checked this on the RBF kernel, we believe directly targeting validation performance is suitable as a baseline and does not hurt performance. Moreover, the baseline we mostly want to compare is to strong Bayesian deep learning methods displayed in the first three columns. For this subsection, we wanted to see if NNGPs are reasonably calibrated for the pure regression tasks since most of empirical evaluation in the paper is focused on image classification.
> >
> >
> > Arora et al. (2020) indeed demonstrated superior performance of infinite-width networks for UCI classification. However, note that their study is on classification using an NTK-SVM and does not consider NNGP nor the calibration of the classifiers. Moreover, our choice of UCI dataset is based on a widely used benchmark in the Bayesian Deep Learning community started from Hernández-Lobato & Adams (2015) and benchmarked in deep ensembles (Lakshminarayanan et al., 2017).
> >
> >
> > **[Bradshaw et al. 2017]**: Motivated by the reviewer's suggestion, we added an additional comparison to RBF-LL which is directly comparable to our NNGP-LL method in Table 3. Note that Bradshaw et al., (2017) trains hybrid GN+DNN model end-to-end which is slightly different from our proposal in Section 5. We added the reference as motivating work for our method.
> >
> >
> >
> > Minor points:
> >
> > We have made edits to clarify our meaning per the reviewer's suggestions.
> >
> > **On disambiguating prior vs inference**: We believe given network architecture and weight initialization distribution, we have exact prior for our BNN. With infinite-width limit, NNGP allows us to capture the functional prior exactly irrespective of what Inference one hopes to use. Moreover NNGP allows exact Bayesian Inference, if one is interested in that, through NNGP for squared loss.
> >
> >
> > Please let us know if there are concerns we have not fully addressed yet.

---

### Official Review · AnonReviewer1 · 2020-10-29
**very clear and interesting experimental evaluation of Gaussian process models with infinite-width neural network kernels.**

**Rating:** 7
**Confidence:** 4

**Review:**

update 1: I thank the authors for the rebuttal. My questions and concerns were appropriately addressed. However, other reviewers raised some concerns regarding the experimental set-up that I have missed. Thus, I will keep my score as is -- 7: good paper, accept.

Summary:

The submission studies Gaussian process models with the infinite-width neural network kernels. In particular, through a series of experiments, the paper investigates their uncertainty properties and answers the question “how calibrated are the predictive uncertainties for in-distribution/out-of-distribution inputs?”: i) a comparison between GP classification with the infinite-width neural network kernels and finite width neural network classification was provided to test the calibration, (ii) a study of these kernels on regression tasks where the GP posterior can be obtained exactly, (iii) a study of using GPs with these neural network kernels on features extracted from a pre-trained network, aka, deep kernel learning of Wilson et al with neural network kernels. Overall, the performance of these GP variants is promising and competitive to the best existing methods.


Assessment:

The paper is very clear and is an interesting read. The experiments are well-designed and the comparison to relevant state-of-the-art methods was provided. Whilst the theoretical contribution is on the light side (no new models or algorithms proposed in this submission), the results from the experiments outweighs this weakness and thus make this paper potentially relevant to the large Bayesian deep learning/GP community. I am leaning toward "accept" and willing to increase the score if my questions below could be clarified.

1. This point is rather philosophical. The calibration metrics are great and they can be used for out of distribution detection. Perhaps this is discussed in previous work but it is not clear to me how these scores can be used to compare methods. For log likelihood, Grosse et al pointed out that a difference of xx nats is significant. Can we quantitatively do the same for these metrics, i.e. the ideal classifier would produce xxx, or a useful threshold for practical use is xxx, or the differences between the whisker plots in figures 2 and 3 are significant. Additionally, are these uncertainties calibrated enough for downstream tasks such as active learning, continual learning, bandits/reinforcement learning etc?. At the risk of asking too much for a conference paper, perhaps since this submission is about understanding the uncertainty of these GP models, an experiment or two on this aspect could be included.

2. I think the difference between the models/inference schemes considered here could be made clearer so that readers could get a clear picture of what methods are being compared to what alternatives. My understanding is that there are the following configuration of models/inf methods -- top: inference, left: net width

net width  Bayesian   point (ML/MAP)
---------------------------------------------------
finite               (1)                  (2)

infinite           (3)                  (4)

(1) is hard to compute exactly. Approximate Bayesian is not calibrated enough. Though there are approximate sampling methods that can scale to the network size considered in this submission but these were not considered.
(2) is cheap and thus often used in practice, but not calibrated.
(3) exact solution can be obtained for regression with Gaussian observation noise, but expensive. Approximate inference/sampling is often required. This submission shows this configuration is better calibrated than existing solutions for (1) and (2).
(4) is not considered in the experiments provided. It might be interesting to consider this.

3. Why not directly optimising the marginal likelihood (in the regression case)? If I understand correctly, Lee et al used this for their MNIST regression experiment?

4. It would be interesting to compare the infinite net (GP) performance with wide-net performance or narrow-net performance, to understand what regime of the networks is being used and what causes the performance difference across model/inf method configurations: is that because of the model, or is that because of the inference method?. Matthews et al had some comparisons like this using BNN with HMC and GP regression.

5. clarity: For figure 1 (left) and those in the appendix, it’s very hard to see the overlapping bars, maybe just plot the curves connecting the top of the bars? For figure 2 (middle) the limits of the y-axis could be changed.

---

> ### Author Response · Authors · 2020-11-21
> **Response to Reviewer #1**
>
> We thank the reviewer for constructive and thoughtful feedback! Here, we address a few points made by the reviewer.
>
> (1) We have added a paragraph in the introduction discussing calibration metrics and citations to additional discussion. While evaluating the models we consider on downstream tasks is out of scope for our paper and we have focused on the benchmark of Ovadia et al. (2019), the fact that both NLL and BS are proper scoring rules and we see improvements in these metrics suggests there should be increased performance on some downstream tasks.
>
>
> (2) Thanks for providing a framework of where our study fits into the bigger picture.  In regard to (4), the solution of infinite-width training is a GP (but not a posterior) with closed form solution given by Corollary 1 in Lee et al. (2019). As such, one can compute the (infinitely large) ensemble of such solutions. Our NTK results correspond to such ensembles. We find both the calibration and accuracy of the NTK and the NNGP are quite similar and we place all NTK results in the appendix (see e.g. S2, S8).
>
> We agree that a comparison to finite with BNNs is interesting. While we did not conduct experiments using these models ourselves, there are results from Ovadia et al. (2019) that can be compared against.
>
> (3) **Marginal likelihood**: This is a great question and was something we discussed when conducting our experiments. We settled on using a validation set rather than the marginal likelihood on the training set as it is closer to how a NN’s hyperparameters are optimized in practice and we did not want to hamper their performance by using marginal likelihood. It also makes for a more direct comparison with prior work, specifically the models from Ovadia et al. (2019). Note that Lee et al. (2018) used this validation set approach also.
>
>
> (4) **Width dependence**: This is indeed a research question of great importance! To serve this goal well, it requires a thorough and comprehensive ablation study into the effect of (1) learning rate, (2) finite-width effect, (3) regularization (L2, early stopping), (3) data augmentation, (4) input processing (e.g. ZCA), (5) normalization (batchnorm), etc. This is similar to Lee et al. (2020) whose main focus is **accuracy** rather than **calibration**. Since this requires a great amount of labor and computation resources, we leave it to future work. However, our results do compare exact Bayesian inference (i.e. NNGP regression) and an “infinite ensemble” under gradient flow (i.e. NTK with MSE loss) in the infinite width setting. Their performance is qualitatively similar. We also point out that width was treated as a hyperparameter for finite NNs in our experiments, and so was optimized over.
>
>
> (5) We have used the reviewer’s suggestion for Fig. 1 to improve its legibility. For Fig. 2, we intended to use these limits in comparison to Ovadia et al., (2019). However we agree, the intent is not clear without explicitly mentioning that and have changed the axes.

---

### Official Review · AnonReviewer2 · 2020-10-31
**Novel contribution with good results, but paper lacks clarity**

**Rating:** 6
**Confidence:** 3

**Review:**

Update: the authors have greatly improved the figures and tables, and expanded the captions, removing my major concern about clarity. I have improved my rating. I not not share the objection of reviewers 5 and 4 about the small size of the CNN, resulting in an inferior baseline in section 3, as the approach is still impractical and mostly a proof of concept that should encourage further research. What matters is that improvements in section 5 are built upon a SOTA baseline, inspired by section 3.

The stated goal of this paper is very ambitious: how NNGPs provide better confidence prediction, in terms of calibration, OOD data  and distributional shift.

After an abstract that is somehow confusing, Introduction and Background  are clear and comprehensive.
Next the authors “describe” their NNGP and try an impressive number of architectures over several datasets, however, assembling the paper seems to have been an hasty business where many clarity issues have been left unresolved, and these are severe, especially for reader such as myself who are not an expert in GP.
The key innovation presented in section 5: the NNGP is just added as a calibration layer on top of a pretrained NN.  It could be of high significance for a practitioner like myself interested in added methods improving the calibration of an existing model, but the authors set aside computational complexity issues and do not give an implementation. There are also serious clarity issues.
Results look excellent, but in the way they are reported in figures and tables, there are too many questions, inconsistencies and readability issues to be sure.

In summary, this seems to be an unfinished write-up where the rich material needs to be better organized with a clear presentation flow, and tables and figures need serious improvements. It has the potential for an excellent publication, proposing a novel solution to an important problem, with rich technical and experimental material.

Detailed comments:

For a reader such as myself who is not an expert in GP but interested in their application to calibration and OOD data, the general introduction in Eq.(1) and (2) makes no sense. I was not even able to parse them and I had to read section 2.4 of (Lee et al 2018) to fully understand how the posterior on a test point is computed. Further down, the authors mention in passing  two critical issues that make NNGP so expensive  without proper explanation:
-	Hyperparameter tuning using Vizier: what are the hyperparameters?
-	Computational cost: they mention it is O(N^3) In the number examples because of the Cholesky decomposition (within what operation? Inversion? reference?). This seems extreme as I see an O(N^2) complexity in other work (Lee et al.)
Note that section A in the appendix did not provide more clarity.

Standard RBF method: can you give a reference?

I spent some time looking for experiments that account for dataset shift before realizing ‘fog’ perturbations represent dataset shift. Same issue with the OOD data: it just appears in Figure 1 and table 1 without proper analysis.

Figure 1: the coloring scheme of the left diagram is not readable. Where is the figure showing entropy?

Table 1 is dropped without any description of the metrics and the datasets:
-	The reader is left guessing what the ‘Fog’ corruptions are, as they are neither introduced nor described, except for a mention of “CIFAR fog corruption” in a table caption. One later learns those are the same as Ovadia shift levels.
-	What is “accuracy”? The reported number for CIFAR10 is 0.487: is this 48.7%? SOTA is more than 90%.
-	Are SVHN and CIFAR100 introduced as OOD? Define mean confidence and entropy? No analysis of results here.

Figure 2 is not readable or comparable to Ovadia et al and should be made larger (maybe remove the Brier Score?):
-	Like in table 1, accuracy is in the 0.6 range when it should be 0.9
-	ECE should range between 0 and .2, not 0 and .8

Table 3:
-	what is the number of examples in NNGP-LL last column? 50K?
-	I have never seen quartiles used for comparison and I do not understand why they are used, as the authors do not seem to exploit the added information. They should be either explained/exploited or removed (recommended to make space for better explanations elsewhere)


Appendix: lots of good material there, some of which could be moved to the main paper (and some material in the main paper should be moved to the appendix, in particular some rows in table 1 and 3)

---

> ### Author Response · Authors · 2020-11-21
> **Response to Reviewer #2**
>
> We are very grateful to the reviewer for the constructive feedback. We appreciate the detailed issues raised regarding the clarity of our manuscript. We have made significant changes, added many more details to our paper, and improved the legibility of the figures.
>
> **Sec. 3**: We have improved the equations and their explanation that introduces the NNGP-C model. We also added additional details on the computational complexity and hyperparameters of the method. The running time is at least cubic in the dataset size due to the Cholesky decomposition of the kernel matrix, which is required to sample from the prior. Note this cubic cost is also necessary for GPs in regression tasks, and indeed this is also true of Lee et al. (2018).
>
> **Fig. 1 and Table 1**: We now introduce the OOD data and fog corruption with a citation in the main text as well as in Fig. 1’s legend. Moreover, the metrics in Table 1 are now introduced in the introduction with more context. The figure showing entropy on the OOD data is in the supplement, which we have now added a reference to. We also improved the presentation of Fig. 1 to make it clearer.
>
> **Fig. 2**: We have improved the legibility of Fig. 2 by changing the y-axis scale as the reviewer asked.
>
> **Table 3**: We are using a 45k/5k/10k train/valid/test split and the last column indicates the full dataset of 45k. Our usage of the quartiles was motivated by Ovadia et. al (2019) (see SM Section G). We are hoping these provide complementary information to the box plots and serve as an easy reference to compare numerical values. Intuitively, the information is similar to what the box-plot conveys, showing how performance and calibration degrades considering all corruptions.
>
> **NNGP-LL implementation and complexity**: We agree that this method could be of use for practitioners! We have added details on the implementation and computational complexity when the method is introduced in Sec. 5 and the SM.
>
> **Appendix**: With the extra page allowance, we have rearranged some material from the appendix into the main text as suggested.

---

### Official Review · AnonReviewer5 · 2020-11-03

**Rating:** 5
**Confidence:** 4

**Review:**

This is an interesting paper which evaluates the calibration of NNGPs, including around OOD detection.

Major points:
1. It is worth noting that in the appendix for arXiv:1808.05587, they explicitly use the full categorical log-likelihood with softmax probabilities (albeit with inducing points), and they find that the resulting model is well-calibrated.  I would be surprised if the use of the Categorical-softmax did not appear in other papers.

2. I am a bit confused about the notion of an "ensemble" of NNGPs.  To my mind, that would imply a number of different NNGPs with different hyperparameters (e.g. depth).  I'm not sure what is being done here, but it appears to be equivalent to a single NNGP?

3. However, my primary concern is around the strength of the NN baselines. They find that the performance of their CNNs and C-NNGPs with CIFAR-10 is very similar, with around 60% accuracy.  For C-NNGPs,  60% accuracy is pretty standard, but for NNGPs, it is really, really bad: with very little effort it is possible to get 90% accuracy, and with a little further tuning, you can get to 95%.  Even when trying to do a fair comparison to infinite networks, other researchers have found 80% accuracy in finite networks (arXiv:1810.05148).  This poor performance brings into question the relevance of their results.  At the very least, the paper needs:
    * A detailed description of why the NN's performance is so poor.
    *Additional comparisons against stronger NN baselines.

4. It isn't clear what the infinite width layers give you that's better than either using Bayesian linear regression over the output weights, or deep kernel learning.

Minor points:
1. Figure 1 (left) is completely unreadable.  Either use separate plots, or use a line (rather than an area) to denote the density.
2. Quote NLL per datapoint, not for full dataset.
3. Exactly what is in table 3 is very unclear: does it aggregate over corruptions?  What are 1k etc. (all of this should be in the caption).
4. Table S1. "confidence on a specific test point being 1.0 to machine precision".  The entropy in that case is zero, and any standard, numerically stable implementation should give that.

---

> ### Author Response · Authors · 2020-11-21
> **Response to Reviewer #5 (part 1)**
>
> We thank the reviewer for the valuable comments and suggestions. We will address the major points addressed by the reviewer first.
>
> (1) Thank you for bringing the analysis to our attention! While we do reference Garriga-Alonso et al. (2019), we indeed missed the initial calibration analysis of the softmax probabilities in the appendix. We updated our draft to address this prior study more clearly. However, we would like to highlight that 1) we are performing evaluation on more complex dataset (natural images versus digits), 2) we use elliptical slice sampling to run asymptotically exact inference and 3) the main goal of Sec. 3 is to compare with a finite network’s calibration for a simple architecture. In our literature review, we found that binary classification is quite widely studied, but references on calibration for multi-class GPs were harder to find, let alone ones using the NNGP.
>
>
> (2) To be clear, we do not consider an **ensemble** of NNGPs. Rather we compare ensembles of finite networks (over multiple initializations) and infinite width networks to predictions of the posterior of the NNGPs. Since the NNGP encodes a distribution over function space, some have referred to this distribution as “an infinite-ensemble.” Since this terminology is clearly confusing, we have removed it.  If there are any other confusing sentences on this point, please kindly let us know so that we can clarify.
>
>
> (3) First of all, when the reviewer mentions
> >"For C-NNGPs, 60% accuracy is pretty standard, but for NNGPs, it is really, really bad:",
>
> we take it to mean that for
> > "finite CNNs, it is really, really bad"
>
> tl;dr: we could not make **NNGP-C** work on the whole cifar-10 on performant architectures (e.g. myrtle) due to poor conditioning of the kernel matrix and the dependence of ESS performance on kernel hyperparameters. We complement our results with *NNGP-R* on such myrtle neworks (85% accuracy).
>
> We agree with the reviewer that there are very well crafted architectures (BatchNorm, Residual / DenseNet, pooling etc) with various training techniques (Large LR, LR decay, Data Augmentation, softmax-cross entropy, L2 regularization, etc) that achieve 90+% accuracy on CIFAR-10 classification tasks.
>
> It is certainly true that one could get a very high performant finite CNN model on CIFAR-10, but this is both 1) architecture and 2) training method dependent. In Lee et al (2020), proper comparison between finite vs infinite networks are discussed and we are following the 'base' configuration SGD training without regularization/augmentation/large learning rate.
>
>  In arXiv:1810.05148, the finite network training uses large learning rate, L2 regularization, Adam optimizer and allows underfitting to achieve 82% accuracy on CNN-VEC. In Lee et al 2020, with a more careful controlled comparison, the authors find that finite CNN-VEC performance is not that different from the kernel’s performance.
>
> Lastly, we would like to emphasize that the infinite-width networks for high-performance models and training techniques are either under-developed yet or computationally challenging to evaluate on a full CIFAR-10 dataset scale.
>
> We would also like to point out that getting NNGP-C to work for the whole CIFAR10 dataset is very challenging as it requires significant tuning even for better conditioned kernels like CNN-VEC. Without tuning we were able to get around 69% accuracy on CIFAR10 with the Myrtle kernel with ESS using only 10,000 training points, but with the same hyperparameters and the full 45,000 training points, the ESS algorithm produced essentially random predictions. We suspect this is due to the poor conditioning of the kernel matrix.
>
> Per the reviewer's suggestion, we updated the draft to clearly discuss this issue. Moreover, we are adding the Myrtle-NNGP architecture studied in Shankar et al., (2020) to the supplementary materials. With this convolutional kernel and NNGP-R, we achieve 85% accuracy on a clean-test set, which is more competitive with well trained CNN models without data-augmentation.

---

> > ### Author Response · Authors · 2020-11-21
> > **Response to Reviewer #5 (part 2)**
> >
> > Continued from (part 1)
> >
> > (4) **[About last layer bayesian linear regression and deep kernels]** Thank you for highlighting last layer Bayesian linear regression / kernels.  The Bayesian linear regression is a very intuitive baseline and we are happy to make comparisons to it.
> >  We did not include it in the original submission because it is much weaker than other gold standard baselines e.g. deep ensembles (see table below). Though our simple FC NNGP-LL (i.e. using infinite-width FCN) is similar to classical deep kernel methods (e.g. RBF, Laplacian kernel), our main innovation here is the usage of the **spatially-aware** kernels: CNN with average pooling and self-attention architectures. Unlike classical kernels that are applied to the **vectorized** features (which is permutation invariant), our CNN-GAP/attention kernels are applied to the **un-vectorized** features and able to explore their spatial structure. As summarized in the table below (also Fig. 3), these spatially-aware kernels outperform classical kernels and finite width network ensembles in both accuracy and calibration.
> >
> > |                                             | Ensemble      | linear GP     | FC-NNGP        | C-NNGP         | Attention-GP   |
> > |---------------------------------------------|---------------|---------------|----------------|----------------|----------------|
> > | Accuracy  (Clean / top 50% all corruptions) | 94.0 / 81.2   | 90.3 / 76.3   | 93.4 / 80.2    | 94.0 / 82.4    | 94.0 / 80.8    |
> > | ECE (Clean / top 50% all corruptions)       | 0.048 / 0.154 | 0.012 / 0.041 | 0.0082 / 0.070 | 0.0085 / 0.033 | 0.0077 / 0.040 |
> > | Brier (Clean / top 50% all corruptions)     | 0.105 / 0.331 | 0.145 / 0.337 | 0.103/ 0.293   |  0.091 / 0.250 | 0.091 / 0.258  |
> >
> > Note that linear regression is a very weak baseline, while more powerful kernels such as Attention-GP/ CNN-GAP-GP are much more expressive, better results than deep kernel / FC-NNGP kernel (see updated figure 3). Furthermore, note that we compare our methods to the current SotA baseline, deep ensembles (Lakshminarayanan et al. 2016), see figure 3.
> >
> >
> >
> > Here are our response for the minor points:
> >
> > (1) **Figure1**: See updated draft for improved version.
> >
> > (2) **NLL per point vs dataset**: Could the reviewer elaborate on this point? In most of the references (e.g. Ovadia et al., (2019) or Lakshminarayanan et al., (2017)) NLL is evaluated on the full dataset as a calibration metric. As far as we can tell, we are following the standard practice but would like to know if we are missing something.
> >
> > (3) **Table 3 clarification**: Here we followed Ovadia et al., (2019), where the table reports quartiles of metrics where quartiles are computed over all corrupted variants of the dataset. The numbers indicate training subset size used, which indicates the effectiveness (and data efficiency) of the NNGP-LL method. While this is mentioned in the end of Section 5, we agree with the reviewer that this should be clearly mentioned in the captions.
> >
> > (4) **NaN in S1**: Thank you for pointing this out. There was a numerical instability that is now fixed.
> >
> > Please let us know if there are concerns we have not fully addressed yet.

---

> > > ### Comment · AnonReviewer5 · 2020-11-24
> > >
> > > Thanks, the response addresses some of my concerns and I have increased my score accordingly.

---

### Author Response · Authors · 2020-11-25
**Summary of changes to the submission**

We thank all reviewers for constructive and thoughtful feedback. We summarize our major changes here.

(1) To address reviewers 4 and 5’s concerns about low performance of CNN-VEC; We add discussion about the choice of the architecture and challenges of applying elliptical slice sampling for myrtle kernel.  We add results of all CIFAR10 corruptions for the Myrtle kernel regression (85% acc) and its finite width counterparts; see Sec. F.

(2) We add the new baseline `Bayesian linear regression last layer`, which is much weaker than the NNGP-LL, which is weaker than the `spatially-aware` NNGP-LL (e.g. CNN-GP-LL); see e.g. Figure 3.

(3) As per suggestion by reviewer 2, we refine our presentation to improve the clarity; add a new section to discuss computation time (complexity and GPU hours) in applying NNGP-LL. The NNGP kernel is very efficient to compute (within a minute for CIFAR10). The more expensive CNN-GAP/Attention kernel (last layer) takes about 1.6 GPU hours.

Please see the detailed response below.

---

### Decision · Program_Chairs · 2021-01-07
**Final Decision**

**Decision:**

Accept (Poster)

**Comment:**

This paper presents an empirical study focusing on Bayesian inference on NNGP - a Gaussian process where the kernel is defined by taking the width of a Bayesian neural network (BNN) to the infinity limit. The baselines include a finite width BNN with the same architecture, and a proposed GP-BNN hybrid (NNGP-LL) which is similar to GPDNN and deep kernel learning except that the last-layer GP has its kernel defined by the width-limit kernel. Experiments are performed on both regression and classification tasks, with a focus on OOD data. Results show that NNGP can obtain competitive results comparing to their BNN counterpart, and results on the proposed  NNGP-LL approach provides promising supports on the hybrid design as to combine the best from both GP and deep learning fields.

Although the proposed approach is a natural extension of the recent line of work on GP-BNN correspondence, reviewers agreed that the paper presented a good set of empirical studies, and the NNGP-LL approach, evaluated in section 5 with SOTA deep learning architectures, provides a promising direction of future for scalable uncertainty estimation. This is the main reason that leads to my decision on acceptance.

Concerns on section 3's results on under-performing CNN & NNGP results on CIFAR-10 has been raised, which hinders the significance of the results there (since they are way too far from expected CNN accuracy). The compromise for model architecture in order to enable NNGP posterior sampling is understandable, although this does raise questions about the robustness of posterior inference for NNGP in large architectures.